# Dendritic Cells in Cancer Immunology and Immunotherapy

**DOI:** 10.3390/cancers16050981

**Published:** 2024-02-28

**Authors:** Laura Hato, Angel Vizcay, Iñaki Eguren, José L. Pérez-Gracia, Javier Rodríguez, Jaime Gállego Pérez-Larraya, Pablo Sarobe, Susana Inogés, Ascensión López Díaz de Cerio, Marta Santisteban

**Affiliations:** 1Immunology, Riberalab, 03203 Alicante, Spain; laura.hato@gmail.com; 2Medical Oncology, Clínica Universidad de Navarra, 31008 Pamplona, Spain; avizcay@unav.es (A.V.); ieguren@unav.es (I.E.); jlgracia@unav.es (J.L.P.-G.);; 3IdiSNA, Instituto de Investigación Sanitaria de Navarra, 31008 Pamplona, Spain; 4Neurology, Clínica Universidad de Navarra, 31008 Pamplona, Spain; 5Program of Immunology and Immunotherapy, Centro de Investigación Médica Aplicada (CIMA), Universidad de Navarra, 31008 Pamplona, Spain; 6CIBEREHD, 31008 Pamplona, Spain; 7Cell Therapy Unit, Program of Immunology and Immunotherapy, Clínica Universidad de Navarra, 31008 Pamplona, Spain

**Keywords:** dendritic cell, vaccine, cancer, immunotherapy

## Abstract

**Simple Summary:**

Although immune check point inhibitors have been established as a new paradigm in cancer treatment, they have shown no clinical benefit in immune excluded or dessert tumors. Dendritic cells promote and coordinate the immune system. Dendritic cell vaccination enriches tumor milieu and potentiates patient systemic antitumoral responses. In this work we review the strengths and weaknesses of dendritic cell vaccines in different solid tumors regarding their role to improve their clinical efficacy. To date, dendritic cell vaccines have induced immune responses in patients without a significant impact on outcome. Improvements in vaccine formulations, selection of patients and combinations with other antitumoral therapies are needed to increase patients’ survival.

**Abstract:**

Cancer immunotherapy modulates the immune system, overcomes immune escape and stimulates immune defenses against tumors. Dendritic cells (DCs) are professional promoters of immune responses against tumor antigens with the outstanding ability to coordinate the innate and adaptive immune systems. Evidence suggests that there is a decrease in both the number and function of DCs in cancer patients. Therefore, they represent a strong scaffold for therapeutic interventions. DC vaccination (DCV) is safe, and the antitumoral responses induced are well established in solid tumors. Although the addition of checkpoint inhibitors (CPIs) to chemotherapy has provided new options in the treatment of cancer, they have shown no clinical benefit in immune desert tumors or in those tumors with dysfunctional or exhausted T-cells. In this way, DC-based therapy has demonstrated the ability to modify the tumor microenvironment for immune enriched tumors and to potentiate systemic host immune responses as an active approach to treating cancer patients. Application of DCV in cancer seeks to obtain long-term antitumor responses through an improved T-cell priming by enhancing previous or generating de novo immune responses. To date, DCV has induced immune responses in the peripheral blood of patients without a significant clinical impact on outcome. Thus, improvements in vaccines formulations, selection of patients based on biomarkers and combinations with other antitumoral therapies are needed to enhance patient survival. In this work, we review the role of DCV in different solid tumors with their strengths and weaknesses, and we finally mention new trends to improve the efficacy of this immune strategy.

## 1. Introduction

Patient characteristics (such as age, genetics, microbiome, previous infections or exposure to immune modified drugs), as well as tumor features—tumor mutational burden (TMB), microsatellite instability (MSI), mismatch repair deficiency (dMMR), specific immune signatures such as interferon or APOBEC and tumor microenvironment (TME) composition and function (tumor infiltrating lymphocytes (TILs))—determine an inherent immunological status in cancer patients, as well as their response to immunotherapy [1,2,3]. Dendritic cells (DCs) play a crucial role in the immune systems’ ability to recognize and eliminate tumor cells. These specialized immune cells have the ability to coordinate both the innate and adaptive immune systems. As part of the innate system, DCs produce protective cytokines (interleukin (IL)-6, IL-12) and growth factors in response to “danger” signals that modulate ongoing immune responses. As antigen-presenting cells (APCs), DCs internalize tumor antigens that are released into the TME. After DC maturation, they efficiently present these peptides through the major histocompatibility complex (MHC) molecules MHC-I and MHC-II to naïve T-cells in the lymphoid tissues, inducing adaptive responses mediated by CD4+ and CD8+ T-cells. By interacting with DCs, naïve T-cells differentiate into effector T-cells with several functions, which ultimately results in the generation of a tumor-specific cellular and humoral response. In addition, DCs interact with other cells of the innate immune system, such as natural killers (NKs), macrophages or mast cells, resulting in the generation of a powerful and complete immune response [4,5].

Evidence suggests that there is a decrease in both the number and function of DCs in cancer patients, with a deeper decrease in metastatic tumors than localized tumors [6,7,8]. Moreover, the powerful immunosuppressive environment generated by tumors interferes with antigen presentation, maturation and the proper function of DCs via several mechanisms, avoiding an effective tumor-specific immune response [3]. Furthermore, lack of immunogenicity in some “cold” tumors based on the absence of T-cell infiltration or due to a dysfunctional or exhausted T-cell status in the inflamed tumors could be recovered through DCV. 

However, it has been documented that it is possible to generate DCs in vitro from cancer patients or to isolate DCs from peripheral blood (natural DCs) and modify them to make them more competent. The use of these DCs, generated and pulsed with tumor antigens to induce an immune response, has been proposed as a therapeutic strategy for certain tumors [4,5]. The aim of these DC-based vaccines is to stimulate the patient’s own immune system to conduct an antitumoral response that eliminates the malignant cells. In addition, this response can generate immunological memory which may prevent relapses of the disease.

The pioneering clinical studies with DC vaccines (DCV) were carried out by Ron Levy in the treatment of B-cell lymphoma [9] and by Frank Nestle in melanomas [10,11]. Since then, numerous clinical trials have been carried out with this strategy. Despite the promising initial results, the findings of subsequent clinical trials were not satisfactory and the interest in this approach decreased. There are many reasons that may have contributed to the limited success of this strategy in some clinical trials [12,13,14,15,16,17,18] that are described below.

### 1.1. An Inadequate Formulation of the Vaccine

The use of immature DCs which are poorly immunostimulated may not have generated an effective antitumor immune response, leading to unsatisfactory results in clinical trials. This is due to the fact that immature DCs have low surface expression of chemokine receptors and co-stimulatory molecules, and they are not able to produce cytokines that are mandatory for T-cell activation. However, mature DCs express higher levels of both MHC-II and co-stimulatory molecules, generating a more effective and powerful immune response based on antigen presentation and migration to lymph nodes. Differentiated T-cells produce inflammatory cytokines that translate into activation of both CD4 and CD8 effector T-cells [19].

### 1.2. A Poor Choice of the Antigens

The use of a single tumor antigen or poorly immunogenic antigens leads to a poor immunogenic DCV formulation. Depending on the source of the tumor antigens employed, the resulting immune response may vary. Therefore, DCs must be loaded with the relevant tumor antigens. Peptides from tumor antigens, RNA-encoding tumor antigens, proteins, whole tumor cells or tumor lysates have been mainly used for the loading of DCs. They all have advantages and disadvantages, and currently, there are no data demonstrating the superiority of any of them, although it seems clear that loading DCs with MHC-I and II epitopes could promote the quality of the immune response generated. Undoubtedly, the most interesting option is to use neoantigens [20], which are tumor-specific proteins that arise from somatic mutations in the tumor. However, this requires its characterization, and this is a costly and time-consuming process. Therefore, another very attractive option is to use material from autologous tumor lysates, which are a good source of personalized neoantigens that avoids the need for identification [21]. This material has many advantages because it contains epitopes for cytotoxic CD8 T and for CD4 helper T-cells, which are essential for generating a potent cytotoxic and memory T-cell response. In addition, the use of this antigenic source reduces the risk of escape of tumor variants.

### 1.3. The Route of Administration of the Vaccine

Although there is no consensus establishing the best route for DCV administration, the quality of responses may be different. For example, the intravenous administration used in some clinical trials may not be the most suitable for achieving the migration of DCs to the lymph nodes where they induce the immune response. The oral route is related to anatomical barriers, such as the gastrointestinal tract, the pH acid milieu and digestive enzymes that hinder the antigen presentation to APC and T-cell activation [22]. Regarding other common routes, intradermal administration has shown a better antitumor T-cell induction when compared with intranodal vaccination [23].

### 1.4. Inappropriate Selection of Patients

Some of the negative results of DCV may also be due to the trend in early-phase clinical trials of including patients with advanced or heavily pretreated metastatic disease. These situations are linked to a tumor-associated immunosuppression. However, therapy-naive oncologic patients with a strong immune system may respond better to antitumoral strategies. In the same way, polypharmacy or patient comorbidities could worsen antitumoral responses. 

Due to these disadvantages, clinical progress in DCV stalled for years, until enthusiasm was reignited by the FDA approval of Sipuleucel-T (Provenge®) for the treatment of prostate cancer in 2010. Sipuleucel-T had no impact on progression-free survival (PFS), but the improvement in overall survival (OS) was clear among men with metastatic castration-resistant prostate cancer [24]. 

Since then, there has been a renewed interest in the development of clinical trials with DCV, demonstrating that the field of DC-active cancer immunotherapy remains attractive. Most of the clinical trials performed with DCs in cancer patients have used DCs derived from monocytes and obtained after culture with IL-4 and granulocyte–macrophage colony-stimulating factor (GM-CSF) for their differentiation, as well as several cytokine cocktails for their maturation. In this respect, a widely used maturation cocktail has been the one composed of tumor necrosis factor-α (TNF-α), IL-1β, polyinosinic:polycytidylic acid (poly-I:C), IFN-α and IFN-γ, generating mature α−type-1 polarized DCs (αDC1) [25,26,27]. These cells produce higher levels of IL-12p70, a molecule necessary for priming a TH1 response, than those generated with the previous protocol (TNF- α, IL-1β, IL-6 and prostaglandin E2 (PGE2)) [25,26,27]. Moreover, when pulsed with tumor antigens, αDC1 induce a potent cytotoxic T-lymphocyte response against the tumor.

However, it is interesting to consider alternative DC sources (Figure 1). Two main types of DCs can be distinguished in peripheral blood: plasmacytoid DCs (pDCs) and myeloid DCs (mDCs). The pDCs are specialized in the recognition of viral antigens, since they lack expression of Toll-like extracellular receptors (TLR) and are the primary producers of type-I interferons (IFN), especially after TLR7 and TLR9 activation [28,29]. They are located mainly in T-cell areas of the lymph nodes and express BDCA2 and BDCA4, but not CD11c [30,31,32]. In the context of cancer, pDCs seem to exhibit mainly tolerogenic properties and are associated with a negative prognosis. However, when appropriately activated, they develop the capacity for cross-presentation and thus become powerful activators of antitumor responses [33,34,35,36].

Myeloid DCs are specialized in immunity against bacteria and fungi. They are located mainly in the marginal zone of lymph nodes and express MHC class II and CD11c. These cells express extracellular TLRs (TLR1, TLR2, TLR4-6) and endosomal TLRs (TLR3 and TLR8), which are responsible for the ability of mDCs to produce IL-12, thereby inducing a Th1 response [29]. The mDCs population is subdivided into two classes, based on surface expression of BDCA1/CD1c or BDCA3/CD141 [29,37].

BDCA3+ DCs are especially important in inducing antitumor responses, and due to their ability to cross-present tumor antigens to CD8+ T-cells [38,39,40,41,42], these cells have been termed cross-presenting DCs. They express MHC class II, CD141+ (or BDCA3+), XCR-1+ and CLEC9A+ that allows detection of damaged cells via binding to exposed actin filaments [43,44]. Therefore, these cells are specialized in the detection and uptake of necrotic cells, and as mentioned above, they stand out in the cross-presentation of these antigens to T-cells [38,45,46]. Cross-presentation is the presentation of extracellular antigens (which are usually bound to MHC-II molecules) to MHC-I molecules: this process allows the activation of CD8+ T-cells, which are essential in the antitumor response. 

DCV has been used clinically for decades, with hundreds of clinical trials testing its usefulness in cancer treatment, and many lessons have been learned from this. Although the published data show evidence that it is possible to induce an immune response against tumors and that DCV is well tolerated and has a good safety profile, clear therapeutic results were only achieved in less than 15% of patients.

Despite this, DCV still represents a promising strategy for cancer treatment, and further progress should be made in optimizing the use of this treatment (Figure 1). Most likely, the combination of DCV with conventional/new treatments is mandatory to achieve synergistic effects. In this way, the incorporation of the immune checkpoint inhibitors (CPIs) as one of the newest therapeutic strategies has brought a revolution in the treatment of some solid tumors, and there is the potential of combining CPIs with DCV as a useful therapy regimen. 

Therefore, the clinical efficacy of DCV may be improved by optimizing the design of the vaccine formulation as well as testing DCV in combination with other therapies in clinical trials.

## 2. DC Vaccines in Solid Tumors

### 2.1. Breast Cancer

By 1998, Fields had described an important antitumoral role of DCs pulsed with tumor lysates in preclinical breast cancer (BC) models [47] driven by the activation of cytotoxic T lymphocytes (CTLs) [48,49]. Further clinical trials with DCV pulsed with HER2/neu or MUC1-derived peptides or DC fusion with BC cells induced immunological responses [50] in BC patients. Ex vivo models showed that co-culture of DCs from healthy donors with human MCF-7 cell lines previously treated with anthracyclines improved CTL activity [51]. However, clinical benefits observed with fusion cells from BC with DCs in metastatic disease [52] were not seen in other studies [53]. To improve the DCV procedure, an antiFoxP3+ Treg depletion immunotoxin was administered before giving viral modified DCV, improving immunogenicity in advanced BC patients [54]. Another group treated metastatic HER2-positive bladder and BC patients with DCs transduced with an adenoviral vector expressing different HER2 domains, demonstrating clinical benefits in up to one third of the patients, together with immune responses [55].

In a non-randomized clinical trial for early disease, Qi et al. treated non-luminal BC patients with tumor lysate-pulsed DCV, showing benefits in 3-year PFS, although they found no impact on OS, with almost 60% of the patients showing immune delayed hypersensitivity responses [55]. Lowenfeld et al. vaccinated ductal carcinoma in situ (DCIS) and early HER2-positive BC patients before breast surgery with HER2 peptide-pulsed DCV and showed 28% of pathological complete response (pCR) in DCIS (only 8.5% in BC), with immunogenic activity in the sentinel lymph nodes as well as in the systemic blood, with no adverse effects [56]. Our group reported that DCV showed an increased immune response in the tumor (higher pCR), its milieu (a rise in TIL and PD-L1 expression) and the peripheral blood (phenotypic changes in immune cells, increased lymphocyte proliferation, IFN-γ production and humoral responses) in paired samples. However, no dramatic changes in survival were found in non-overexpressing HER2 early BC patients [57,58].

On the other hand, triple negative (TN) tumors have a grim prognosis but are also known to respond better to immunotherapy than other BC biologic subtypes based on higher TIL levels, more PD-L1/PD-1 activation, increased TMB and an increased neoantigen (neoAg) production [2,59,60,61,62,63]. Regarding this issue, early TNBC stands out via increased TIL and PD-L1 expression as compared to more advanced diseases [64]. Although the addition of checkpoint inhibitors (CPIs) to chemotherapy has opened a new landscape in the treatment of TNBC patients in both the early [63,65,66,67,68] and the advanced disease [69,70,71,72], mainly due to a better outcome, they have proved to be of no clinical benefit in immune desert tumors or in those tumors with dysfunctional or T-cell exhausted, excluded or ignored phenotypes [61]. In this way, DCV has demonstrated the ability to modify the tumor microenvironment and to potentiate systemic host immune responses as an active approach to treat BC patients by increasing T-cell infiltration within the tumor [58], as well as by increasing PD-L1 expression in tumoral cells and stimulating systemic T-cell activity against BC [57]. 

Application of DCV in BC seeks to obtain long-term antitumor responses by enhancing intrinsic immunity or by generating de novo immune responses. This could be achieved by infusing mature DCs previously loaded with tumor antigens ex vivo or by targeting neoantigens and adjuvants directly to DCs in vivo [73]. 

DCV presents limited toxicity when used as monotherapy or in combination with radiation or other biologic therapies in BC. Combination of DCV with chemotherapeutic agents such as cyclophosphamide, gemcitabine or temozolamide [74] increases toxicity but also shows improved clinical responses via mechanisms such as triggering immunogenic cell death, modifying the permeability of CD8 T-cells to cytolytic factors and decreasing Tregs and myeloid-derived suppressor cells (MDSCs). Radiation therapy upregulates class I MHC and enhances tumor cross-presentation in BC [75].

In the same way, DCV upregulates expression of the PD-1/PD-L1 axis as an adaptive resistance mechanism due to an increased infiltration of TILs within breast tumors [76]. In fact, this transformation from cold into hot immune tumors could increase the target population to benefit from CPIs after DCV together with chemotherapy (Figure 2). Therefore, chemoimmunotherapy combinations may be a more effective therapy for patients. With this strategy, a chemotherapy de-escalation approach may be used to avoid overtreatment of patients and unwanted toxicities. Thus, benefits of immunotherapy are amplified when combined with other antitumoral strategies with an enhancement of T-cell activation, Tregs depletion and reversion of T-cell exhaustion. All these efforts need to be applied in earlier stages of the disease, when the immune system is less affected by tumor growth and works efficiently. 

### 2.2. Brain Tumors

Diffuse gliomas constitute the most frequent malignant primary brain tumors in adults, particularly World Health Organization (WHO) grade 4 glioblastoma (GBM) [77]. Despite recent advances in its molecular characterization and continuous efforts focused on the development of novel therapeutic approaches, the treatment of glioblastoma has remained unchanged since 2005, when maximal safe resection followed by radio-chemotherapy with Temozolomide became the standard of care for patients suffering from this devastating disease [78]. Therapeutic alternatives following recurrence, which typically occurs within 6 to 8 months after diagnosis, are even more scarce. Given the dismal prognosis of these patients, with median overall survival times not exceeding 17 months, novel and more effective treatment options are urgently needed. 

In this context, immunotherapy is still being investigated as a potential therapeutic tool in GBM patients. Among different immunotherapeutic strategies, DCV has been extensively explored in the past decade for such an immunologically “cold tumor”, with promising but not yet consistent results. 

Early studies in animal models of GBM showed that DCV triggered a specific anti-tumor immune response and resulted in antitumoral activity with tumor growth reduction and prolonged survival, leading to the translation of this strategy to the clinic through phase I-II trials. Only a few GMB clinical trials were randomized studies and were designed to achieve a high level of efficacy [79]. Patient populations included in these clinical trials have been heterogeneous, with newly diagnosed, recurrent GBM or even both groups of patients. Most of them included only patients who had undergone extensive tumor resection with minimal residual disease following surgery, but in some cases, patients who only had a biopsy or even no surgery were also included. Apart from the well-known association between survival and extent of surgery, complete or near complete resections might also reduce the local immunosuppressive microenvironment and facilitate the effect of DCs. Other favorable clinical characteristics, such as good performance status and the need to discontinue corticosteroids prior to vaccination, were also frequent inclusion criteria in most studies [80,81]. In most trials conducted on newly diagnosed patients, DCV was administered concomitantly with radiotherapy or chemotherapy. 

A key factor in DCV in cancer therapy relies on the presence of antigenic targets in the tumor cells. It is unknown whether whole-tumor cell sources of TAA (obtained from either tumor lysates, tumor eluted peptides, tumor mRNA, tumor-derived extracellular vesicles or tumoral stem cells) or standard molecularly defined tumor-associated antigens (such as epidermal growth factor receptor variant III) are superior in inducing anti-tumor immune responses and clinical efficacy. However, because of the extreme histopathological and molecular heterogeneity of GBM, vaccines targeting specific GBM-associated antigens are restricted to selected patients. Thus, autologous rather than standardized antigen-based vaccines might ensure a more personalized and better targeting of the full repertoire of antigens present on the patient’s tumor and might avoid mutations of single targeted specific antigens. 

A wide range of doses have been explored in trials with DC vaccination for GBM patients [80]. Starting from that basis, the optimal DC dose is still difficult to define because many parameters, such as the type of DC, the vaccination schedule and the delivery, might influence the anti-tumor immune response and efficacy. The route of application of DCs also varies across different studies, including intranodal, intratumoral, intravenous, subcutaneous and intradermal application. The latter one appears to be the most advantageous and has indeed been the most frequently used [80]. 

Overall, DCV for GBM has shown excellent safety and tolerability profiles in clinical trials. Adverse events are mild and easily manageable. The most frequently observed side effects related to DCV are local reactions in the site of injection, such as itching, erythema and pain. Other reported toxicities are mostly attributable to the disease itself or to other therapies used concomitantly [80,81]. In a non-randomized phase II trial, no relevant adverse events or toxicities attributable to the immunotherapy were registered [82]. Increased levels of IFNγ and other proinflammatory cytokines following DCV as well as antitumoral cytotoxic responses have been described [83], thus reflecting the induction of an immune response in GBM patients treated with DC vaccination. 

Regarding efficacy, non-randomized phase I and II trials suggest a benefit to survival in patients with glioblastoma treated with DC vaccination as compared to matched or historic controls, with a median OS ranging from 15 to up to 40 months for newly diagnosed patients [80]. Among them, our group conducted a phase II trial evaluating the efficacy and safety of autologous DCV in addition to fluorescence-guided surgery and standard radio-chemotherapy in 32 patients with newly diagnosed GBM. Patients additionally received standard treatment with radiation and temozolamide. Median progression-free survival (PFS) was 8.0 months and median OS was 27.4 months, suggesting that autologous DCV is feasible and safe and that the addition of this immunotherapy modality in GBM could improve OS [82]. Randomized phase II trials revealed mixed results [84,85,86,87,88], and a single randomized phase III trial was inconclusive because of its cross-over design, with nearly 90% of patients in the control group receiving DCV after recurrence [89]. A recent *post-hoc* analysis comparing survival of the entire patient population with those from matched external control populations suggested an improvement in median OS (19.3 months versus 16.5 months) [79]. However, major differences in methodological design and enrolled patients among such DCV trials and those whose data were used as external controls for comparison present major limitations, and therefore, valid conclusions cannot be drawn [90,91]. 

### 2.3. Colorectal Cancer

With more than 600,000 deaths estimated each year, colorectal cancer (CRC) is the second leading cause of cancer death in Europe [92]. In the metastatic setting, 5-year survival rates are only 14% with current standard-of-care (SOC) treatments, including surgical resection of primary lesions and metastases and systemic regimens that combine chemotherapy with first-line targeted therapies such as anti-EGFR (epidermal growth factor receptor) or anti-VEGF (vascular endothelial growth factor)-based strategies [93]. Regarding immunotherapy, FDA approval has only been granted for CPIs in patients with the MSI-high or dMMR CRC subtypes [93]. Even in this setting, lack of responses or the appearance of hyperprogression have been well documented [94]. Besides, CPIs have very limited efficacy in MSS mCRC patients, with <5% best overall response with a median OS of 5 months. Regarding DCV, there is great variability regarding loading, with defined peptide sequences following their genomic identification through transfection of total tumor mRNA or with the use of autologous tumor lysates [95]. Results of early-phase clinical trials of DCs pulsed with exogenous tumor-associated peptides such as carcinoembryonic antigen (CEA) or with autologous tumor lysates demonstrated safety and immune-specific responses with no survival benefit [96]. Given the potential synergy of DCV with chemotherapy, a pilot study tested the effect of DCV pulsed with keyhole limpet haemocyanin (KLH) and CEA peptides during adjuvant oxaliplatin/capecitabine chemotherapy in stage III colon cancer patients. Different routes of DCV administration were based on pre-clinical models showing that IV delivery provides a better anti-tumor response against visceral metastases (such as liver, lung, brain), whereas non-visceral metastases (lymph nodes, bone, skin, pleura or peritoneum) respond better to the intradermal route. KLH-specific T-cell responses remained unaffected by the chemotherapy, but the humoral response was impaired. Neither additional toxicity nor disease relapse were reported due to a short follow-up period [97]. 

Regarding metastatic disease, the NCT02503150 trial compared the efficacy and safety of 12 cycles of mFOLFOX6 chemotherapy and DCV with modified chemotherapy alone followed by maintenance with 5-Fluorouracil plus every 3 months DCV or 5-Fluorouracil monotherapy. Another phase II randomized clinical trial compared DCV with best supportive care, with differences in OS in those patients who responded to DCV (but not in the whole cohort) and with a tumor-specific immune response [98]. These results led to the single arm phase I/II trial of avelumab plus autologous dendritic cell vaccine in pre-treated MMR-proficient metastatic CRC patients. Combined therapy was safe and well tolerated, but in the interim analysis, only 11% of the patients were disease-free at 6 months [99]. 

Our group tested DCVs loaded with autologous tumor lysates for their potential to avoid or delay disease relapses in patients with surgically amenable liver metastasis of colon adenocarcinoma treated with neoadjuvant chemotherapy, surgery and adjuvant chemotherapy. Fifteen patients with disease-free resection margins were randomized 1:1 to receive DCV versus no therapy [100]. Even though the number of patients was small, follow-up of the patients indicated a clear tendency to fewer and later relapses in the DCV arm (median disease-free survival 25.2 months versus 9.5 months). Treatment with DCs was safe and only mild side effects were recorded [101]. Although immune responses have been generated following DCV, a limited clinical benefit has been shown in all these trials. Potential areas of improvement include the selection of the most suitable patients, improving identification of more immunogenic TAA to load with, ex vivo culturing techniques and inoculation routes and better selection of synergistic therapies to combine with DCV [102,103], such as oxaliplatin, which activates DCs in a toll-like receptor-4 (TLR4)-dependent manner [104]. One of the key limitations of DCV relies on its inability to overcome immunosuppressive properties of the tumor microenvironment. Therefore, treatment combinations of DCV with prior cyclophosphamide in order to suppress Tregs have been conducted [105]. Other strategies that could enhance DCV efficacy are based on a high expression of calreticulin on tumoral cells that induces phagocytosis and serves as a “find me” signal by DC precursors [106]. 

In preclinical models, DCV achieves higher antigen-specific tumor protection when animals are first sensitized to anti-angiogenic drugs [107]. Sustained tumor-produced VEGF can induce PD-L1 expression on myeloid DCs, impairing mature DC mobility and suppressing expression of MHC class II and other costimulatory molecules [108]. Reported synergism between the VEGF blockade and chemotherapy relies on the “vascular normalization” induced by antiangiogenic drugs, which allows combination schedules to more effectively distribute chemotherapy within the tumor. This blood flow restoration reduces immune suppressive properties like hypoxia, acidosis and downregulation of leukocyte adhesion molecules (e.g., ICAM-1, VCAM-1, E-selectin and CD34) and may increase the ability of DCs to induce antitumor immunity [109]. In summary, although immune response generation following DCV has been shown in all these trials, better combinations with DCV are needed to improve clinical benefit in CRC patients. 

### 2.4. Gynecological Cancers

Uterine, ovarian, cervical, vaginal and vulvar cancer are the five most widespread types of gynecologic cancers. Uterine cancer is the most frequent gynecologic neoplasm, while ovarian cancer (OC) remains the most lethal malignancy, with a 5-year OS around 50% and nearly half of patients with distant disease at diagnosis [110]. Only cervical cancer (CC) has a screening test and could benefit from a preventive human papillomavirus cancer vaccine [111].

Currently, there are multiple indications for CPI based-immunotherapy (anti-PD-1/PD-L1 agents) in this group of malignancies, either combined with chemotherapy in first-line advanced CC for PD-L1-positive tumors, combined with tyrosine kinase inhibitors (TKIs) in second-line metastatic endometrial cancer MMR-proficient patients or as monotherapy with extended approval based on several biomarkers (PD-L1, MMR genes, TMB) [112,113,114,115,116,117,118]. Over the last decade, an increased interest has emerged in immunotherapy research beyond the PD-1/PD-L1 axis, including the application of DC-based therapy. The use of DCs immunotherapy alone to treat gynecological cancers is insufficient due to tumor-induced immunosuppression. The infiltration of immunosuppressive cells in the tumor microenvironment drives DCs into a dysfunctional state by disrupting their immune role and metabolism, resulting in an impairment in antigen-presenting cell (APC) function and number and promoting cancer progression. However, DCVs are safe and effective and provide clinical improvement to these gynecological cancer patients [119]. HPV types 16 and 18 are present in 90% of CC cases [120]. As compared to other high-risk HPV (hrHPV) genotypes, HPV16 and 18 infections are more probable to result in invasive CC in a shorter period of time, making them more oncogenic HPV variations [121].

Despite preventive HPV vaccinations and conventional therapies, CC still has a considerable morbidity and mortality, particularly in premenopausal women. With the FDA’s approval of pembrolizumab in combination with chemotherapy in the treatment of locally advanced and metastatic CC in October 2021, immunotherapy has recently changed the landscape of CC patient management [112]. 

Adoptive and active cell-based therapies have added the concept of individualized immune treatment in which CAR-T, TCR-T and TILs could be employed alone or in combination to eliminate early-stage CC [122]. It has been observed that the concentration of DCs is low in cervical tumors while that of Treg cells is higher, which may be significantly associated with the persistence of hrHPV [122]. Consequently, DCs gradually lose their ability to deliver antigens to tumor cells [123], particularly as a result of cancer cells secreting RANKL. This is a potential immunoevasion strategy for cervical cancer, together with Treg infiltration. DCV works by presenting HPV antigens to innate and adaptive immune cells. In order to increase cell therapy efficacy, DCs and T-cells have been modified ex vivo with small interfering RNAs against immunosuppressive targets and then re-infused back into the patients [124]. A phase I clinical trial was conducted in patients with stage I-II CC who were treated with DCV. These patients were subcutaneously injected with a DCV-carrying keyhole limpet hemocyanin (KLH) and full-length HPV16/18 E7, which produced a CD4+ T-cell and cervical cancer-specific humoral immune response [125].

In OC patients, clinical trials have established the safety of DCV, whereas effectiveness varies depending on production process, delivery and study design. Personalized DCVs have emerged as a hot topic due to the extensive application of next-generation sequencing and bioinformatics analysis in a variety of scientific fields. The discovery of a lower number of DCs in patients with OC than in healthy donors has driven the incorporation of DCV into the clinic [126]. Currently, more than 20 DCV clinical trials in OC looking for new efficient strategies against one of the most aggressive solid tumors have been registered on ClinicalTrials.gov (https://clinicaltrials.gov/search/ovariancancer&Dendritic, accessed on 27 February 2024) [127]. Spisek’s group conducted several phase II trials to determine if the addition of monocyte-derived cell (moDC) vaccines to the SOC therapy could provide additional benefits as compared to SOC alone. The potential benefit of moDC vaccination in addition to SOC was evaluated in the first phase II clinical trial, including stage III-IV platinum-sensitive OC patients relapsing after first-line chemotherapy, with vaccination initiated after two cycles of chemotherapy [128]. In a second phase II clinical trial, stage III OC patients were treated with the moDC vaccine either concurrently with chemotherapy, sequentially, or with chemotherapy alone (SOC), with an improvement in PFS in the sequential arm (HR = 0.39, 95%CI 0.16–0.96; *p* = 0.03) [129]. Furthermore, the outcome of 56 OC patients sensitive to SOC and treated with mucin1 loaded-DCV as maintenance therapy was analyzed in another randomized phase II clinical trial, with positive results based on immunogenicity and safety but with no changes in PFS. An increased PFS and a non-significant 16-month improval in OS were observed in OC patients on a second remission status within a chemotherapy schedule [130].

In most studies, DCV has been reported to prolong tumor PFS, even though the impact on OS has not been significant. Therefore, these findings show that DCV has limited benefits. However, further research is required at the level of the DCV manufacturing, the potential combination of drugs with DCV and/or the predictive/prognostic biomarkers used to select patients to enable further improvement in DCV therapy. Prospective cohort studies with large samples will provide the greatest evidence in the future [19].

### 2.5. Melanoma

Melanoma has historically been considered an immunogenic tumor, and it has been known for the most impressive results with immune CPIs since the early 2010s. Dendritic cells are increasingly considered a key step of a successful immune recognition of tumors, with conventional type-1 DC (Cdc1) having gained the greatest interest in recent years because of their specialized capacity to directly prime CD8+ T-cells [131]. Therefore, various approaches have targeted this immune cell population in melanoma patients [132].

DCV based on peripheral blood-derived DCs pulsed with HLA-matched off-the-shelf peptides or tumor lysates injected into uninvolved lymph nodes were well tolerated, induced peptide-specific immune responses and, more impressively, produced objective tumor responses in 5/16 advanced melanoma patients [133]. A similar clinical trial with MAGE-3A1 peptide-pulsed monocyte-derived DCs also resulted in the expansion of antigen-specific cytotoxic lymphocytes, as well as clinical responses of tumors in 6/11 patients [134]. However, a phase 2 clinical trial that compared the immunogenicity of melanoma-associated peptides administered randomly in a GM-CSF emulsion or loaded into DCs showed a higher capacity for the cell-free preparation in terms of antigen-specific T-cell response induction [135]. Interestingly, a publication in 2005 reported that the presence of antigen-specific lymphocytes in delayed-type hypersensitivity test skin biopsies predicted progression-free survival in patients treated with a variety of DCV protocols [22]. This finding raises the key unanswered question of the factors that influence the induction of an adequate T-cell priming in patients to better identify patients in need of DCV to mount an effective immune response against the tumor.

Recent technological advances have allowed a more individualized approach to antigen selection to broaden antitumor immunity. A phase I trial which included only three patients with resected stage III melanoma was conducted. Personalized MHC-restricted neoantigen peptide-pulsed ex vivo induced DCs were able to amplify existing antigen-specific T-cell responses but could also produce responses to other antigens that were not detectable prior to vaccination [136]. 

Other innovative approaches directly isolated peripheral blood Cdc1 mDCs, loaded them with gp100 and tyrosinase-derived HLA-A*02:01-restricted peptides and administered them into the clinically tumor-free lymph nodes of 14 evaluable advanced melanoma patients [137]. Objective responses of individual metastases were described, although no patient met the overall response criteria. A similar clinical trial by the same group described that Pdc vaccines obtained from peripheral blood were feasible and safe in advanced melanoma patients [35]. A mixed response was described in one out of fifteen evaluable patients, but none obtained an overall tumor response. Other groups have reported that vaccination with natural classic and pDCs is safe and feasible [138]. Another innovative clinical trial selected Langerhans-type DC (CD83+ CD86^bright^HLA-DR^bright^CD14^neg^) obtained ex vivo from CD34+ cells from the peripheral blood of resected stage IIB to IV melanoma patients. Increased clonality, cytokine production and cytotoxicity markers were observed on antigen-specific T-cells, with no clear association with relapse risk [139]. Similar early-phase clinical trials have confirmed that DCV is feasible and safe in combination with chemotherapy [140,141]. Other strategies, such as DC loading with NKT cell agonists [142] or DC activation via CD40L [143] and different antigen loading approaches (melanoma cell and DCs fusion, co-culture and tumor cell lysates), have also been explored [144]. Butterfield et al. developed several clinical trials with DCV in melanoma patients. The first such clinical trials were reported in the early 2000s, and occasional clinical responses were described with DCs pulsed with the MART-1 peptide [145,146]. Two additional trials by the same group explored the loading of melanoma antigens through adenoviral transduction. Epitope spreading against antigens not included in the vaccine was described [147], as well as additional clinical responses in a minority of patients [148]. 

Other innovative antigen-loading approaches such as Mrna electroporation have been attempted. This strategy, with the electroporation of four melanoma-associated antigens linked to an HLA class II targeting signal, was reported to be safe and clinically active in a phase 1b clinical trial [149]. In a phase 2 single arm trial in combination with ipilimumab that included 39 stage III-IV melanoma patients, a remarkable response rate of 38% (8/39 were complete and 7/39 were partial responses) was reported [150], numerically higher than in pivotal clinical trials with ipilimumab monotherapy [151]. Additionally, a retrospective analysis of patients with progressive disease during or after DCV suggested that treatment with antiPD-1 or antiCTLA-4 antibodies or their combination was effective in this setting [152]. In addition, small phase 1 clinical trials have explored the intratumoral administration of mDCs (BDCA1+ and BDCA3+) isolated from peripheral blood in combination with other intratumoral agents such as T-VEC or ipilimumab, with preliminary signs of activity in ICB-refractory patients [153,154].

Overall, the modest clinical activity and the significant variability among DCV protocols have precluded further development of DC-focused therapeutic strategies. Evidence from more advanced clinical trials is very limited in this setting. A phase 2 randomized clinical trial was performed with 144 stage III-IV melanoma patients receiving tumor-lysate-loaded DCV or a placebo in the adjuvant setting [155]. Others reported that disease-free survival at 24 months in the *per-protocol* population was significantly prolonged in patients who received DCV (62.9% vs. 34.8%). However, results from another randomized placebo-controlled phase 3 trial that included 151 patients with the same clinical scenario showed no relapse-free survival benefit from melanoma-associated antigen-loaded DCV [156]. 

### 2.6. Urologic Tumors

Prostate cancer expresses a high number of TAAs, such as prostate-specific membrane antigen (PSMA), prostatic acid phosphatase (PAP), prostate-specific antigen (PSA) or six-transmembrane epithelial antigen of the prostate-1 (STEAP1). Nonetheless, it is one of the few major tumors in which immunotherapy has not become a widely used treatment. 

Yet, prostate cancer was the first tumor in which a DCV was approved for clinical use. Sipuleucel-T (Provenge^®^), a DCV loaded with PA2024, a combination of PAP plus granulocyte–macrophage colony-stimulating factor (GM-CSF), showed increased OS as compared to a placebo in patients with asymptomatic or minimally symptomatic metastatic castration-resistant prostate cancer (Mcrpc) in the IMPACT phase III trial (25.8 vs. 21.7 m; HR = 0.78, 95%CI 0.61–0.98; *p* = 0.03) [24]. Sipuleucel-T has also been explored in the neoadjuvant setting before radical prostatectomy, showing a significant increase in the tumor-immune infiltrate in surgical specimens as compared with pre-treatment biopsies [157]. Sipuleucel-T has been further developed in combination with other agents, such as different sequences of androgen deprivation therapy, abiraterone, ipilimumab or Radium-223 [158]. Despite this compelling data, the clinical use of Sipuleucel-T is low, likely due to its complexity, cost and the availability of alternative treatments in this setting, such as abiraterone or enzalutamide.

Other DCVs under development for prostate cancer include different antigens such as PSMA, MUC1, NY-ESO-1, MAGE and CDCA1, etc., some of which have shown preliminary signals of activity. For example, in a small trial, 21 patients with Mcrpc were randomized to receive a vaccine with DC loaded with recombinant PSMA and survivin peptides or docetaxel and prednisone (SOC). Cohorts showed response rates of 73% with DCV and 45% with chemotherapy [159].

Other DCV approaches have been tested in urothelial tumors in the setting of exploratory phase I-II trials. For example, a vaccination with Wilms tumor (WT1) peptide-pulsed DCs in combination with molecular targeted therapy or chemotherapy induced immunological and clinical responses in patients with NMIBC or renal cancer [160]. 

The relevance of the immune response in renal cancer has been known for decades, and consequently, several vaccines have been developed for this tumor type. Rocapuldencel-T is a DCV combined with amplified tumor RNA plus CD40L RNA, with a mechanism of action similar to Sipuleucel-T. The ADAPT phase III trial compared Rocapuldencel-T plus SOC versus SOC in 462 patients with untreated advanced clear-cell renal cancer [161]. The vaccine induced immunological responses but did not improve OS, although the magnitude of the immune response directly correlated with OS. 

## 3. New Trends

### DCV Pulsed with Neoantigens

Despite important advances in understanding DC biology and its role in cancer, improvement of tools and procedures for DCs preparation and its application to patient therapy, most clinical trials have reported limited results, with promising data on vaccine immunogenicity but poorer efficacy in terms of therapeutic effect, suggesting that there is still room for improvement. In this regard, future perspectives for optimized DCV clinical trials include, among others, selection of the proper DC subset (as previously mentioned for blood-derived BDCA3+ mDCs), improvement of production protocols, identification of relevant antigens to be included in vaccines and combination of vaccines with additional immunotherapeutic approaches. 

Regarding antigen selection, most clinical trials using defined antigens have been based on the use of TAAs, which, despite selective expression in the tumor, are not completely tumor-specific. Central tolerance mechanisms involved in the deletion of T-cells with high-affinity TCRs specific for self-molecules [162] have been proposed as one of the causes explaining the limited efficacy of vaccines based on these antigens. This suggests the necessity of identifying highly specific tumor antigens not subjected to central tolerance mechanisms and capable of inducing high-affinity T-cells. With the development of next generation sequencing (NGS) technologies, sequencing of the tumor genome or exome has become available, and the identification of tumor-specific mutations has grown exponentially over the last several years [163]. These mutations include non-synonymous single-nucleotide variants, indels and other genetic changes that may generate new amino acid sequences. By analyzing the presence of these mutations, the relevance of the tumor mutational burden (TMB) and CPI efficacy has been demonstrated [164,165,166] at the same time that many neoantigens (neoAgs) have been identified, allowing the detection of tumor-specific T-cells and fine characterization of their antitumor properties [167]. Indeed, the relevance of immunity against neoAgs in patients responding to CPI [168], as well as the presence of neoAg-specific T-cells in adoptive transfer products, such as those based on expanded TILs, indicate the importance of T-cells recognizing this type of antigen [169], suggesting that they could be a good target for vaccination protocols. 

Several neoAg-based vaccination clinical trials have been conducted, some of them using DCV [136,170,171,172]. There are some mutations common to different tumors and patients which have been proposed as general targets. In this context, the Nous-209 vaccine encodes a wide selection of sequences arising from common frameshift mutations in tumors with microsatellite instability (MSI) [173]. However, the majority of neoAg-based vaccines have been highly personalized. In most cases, after careful selection, administration of these antigens has demonstrated their strong immunogenicity, and in some cases, clinical effects have been reported [174,175,176]. Although DC vaccines pulsed with tumor lysates contain these antigens, neoAgs are poorly represented in the antigen repertoire displayed by DCs after pulsing, competing for binding to HLA molecules with other less specific tumor antigens and even with highly abundant self-proteins. Therefore, selective formulation of DCV with tumor-specific neoAgs may favor induction of responses against these targets in a better way than current DC vaccines. 

Moreover, in addition to mutation-based neoAgs, other tumor-specific molecules can also be targeted by vaccines. In this regard, current identification of tumor-specific antigens has been carried out by using genomic and transcriptomic data corresponding to those genes encoding classical Mrna molecules that generate proteins annotated in proteomic databases. However, there is a collection of RNA molecules that have not been traditionally considered as a protein source but more recently have shown their capacity to be translated into proteins, generating sequences with potential antigenicity [177]. This protein set, denominated by some authors as the “dark proteome”, may contain proteins specifically translated into tumor cells that could be also considered as tumor-specific antigens [178]. Indeed, studies of the ligandome in tumor cells (the set of peptides presented by HLA molecules) have reported a number of peptides not annotated in protein databases [179], which in some cases may be tumor-specific. This new set includes proteins or peptides encoded via fusion of transcripts, those created by Mrna splicing events, those created by proteasomal splicing or those encoded by long non-coding RNA, small nucleolar RNAs and proteins encoded in ribosomal DNA, as well as peptides with post-translational modifications [178,180]. Crossed analyses of unannotated peptides in ligandome databases with transcriptomic data have revealed that a high number of previously unknown peptides presented by MHC molecules in cancer correspond to transcripts not related to traditional Mrna molecules [181], indicating the relevance of this supplementary source of tumor-specific antigens not considered in current vaccines. 

In addition to improving vaccine intrinsic properties by modifying vaccine characteristics, combination of vaccines with other immunotherapies in the proper setting may provide enhanced response rates. More recently, CPIs, namely those targeting the PD-1/PD-L1 pathway, have become the backbone of most immunotherapy combinations [182]. Their impressive results in an important proportion of hot, highly inflamed tumor types with elevated TMB levels have suggested that promoting tumor inflammation via strategies like vaccines may widen CPI benefit to new patient groups. Furthermore, the immunosuppressive tumor microenvironment has been proposed as one of the causes potentially explaining the poor effect of vaccines despite their immunogenicity. Therefore, release of immunosuppressive brakes imposed on tumor-specific T-cells induced by vaccines would allow responses with stronger functional properties. In this regard, despite the common view of a local effect of CPIs on TILs, there is growing evidence that an important pool of T-cells responding to CPIs correspond to those rescued in the tumor-draining lymph nodes upon antigen presentation by PD-L1-expressing DCs [183]. In this scenario, combination of vaccines (generating a new pool of tumor-specific T-cells) together with CPIs would increase efficacy above that observed by these therapies when applied independently.

## 4. Conclusions

DCVs have been able to induce detectable antitumor immune responses in cancer patients, but the magnitude or the context where this immunity has been primed has not sufficiently impacted clinical outcomes in both the early and advanced scenarios for solid tumors.

New advances in this field will require improvements in the vaccine via selecting more relevant Ags (e.g., neoAgs) or improvements in the route of administration, dose and schedules. This could be achieved through the use of nanomedicine technologies [184] implemented to enhance antigen capture and loading, DC maturation and Ag presentation, resulting in improved delivery to patients. Other potential limitations of DCV are related to the patient’s own immune system and the induction of immunotolerance. 

Moreover, a better selection of patients in randomized phase II-III clinical trials is needed. This includes vaccine administration to appropriate patients using individualized predictive and prognostic biomarkers to identify individuals with tumors lacking TIL or with exhausted or dysfunctional TIL that would benefit from immune-priming strategies. Finally, besides these factors, suitable combinations with already existing therapies have to be explored in order to find synergistic effects between vaccination and CPIs or SOC. This will result in a more effective strategy able to prolong antitumoral effects while maintaining low toxicities [185].

## Figures and Tables

**Figure 1 cancers-16-00981-f001:**
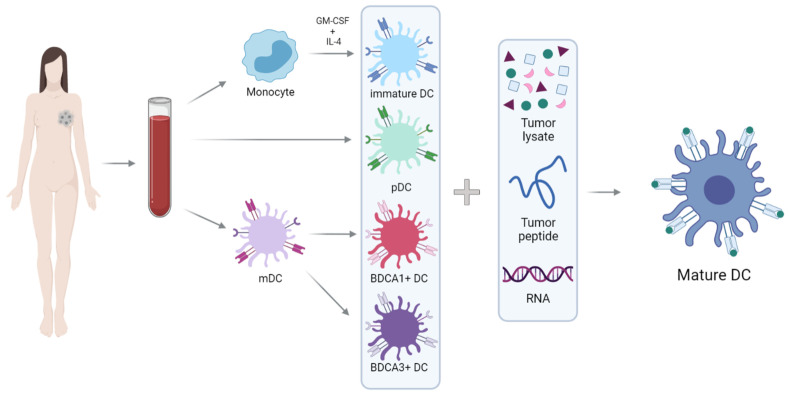
Schematic representation of the generation of DC vaccines. Monocytes isolated from peripheral blood can be differentiated into DCs after culture with GM-CSF and IL-4. Moreover, there are two main types of DCs in peripheral blood: pDCs (plasmacytoid)and mDCs (myeloid). DCs must be loaded with peptides from tumor antigens, RNA-encoding tumor antigens or tumor lysates to obtain the final product: mature and pulsed DCs.

**Figure 2 cancers-16-00981-f002:**
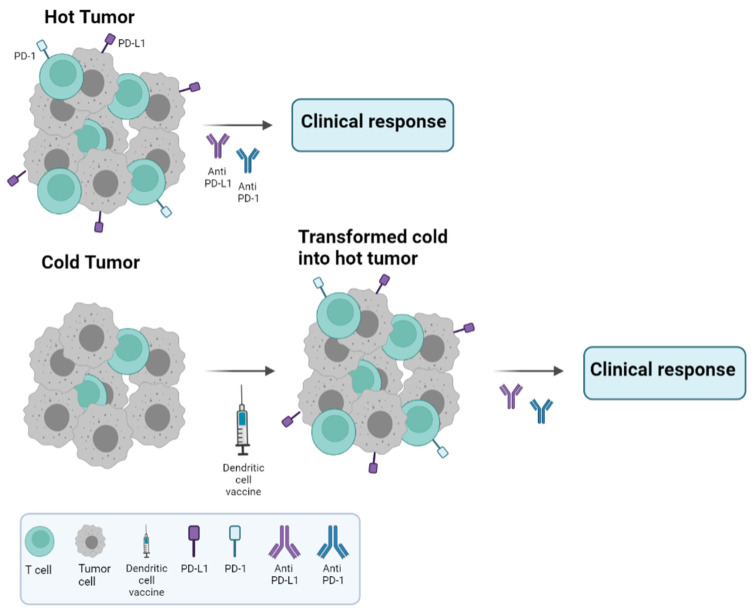
Cold versus hot tumors. Treatment schedule recommended according to the type of tumor. Hot tumors are characterized by high lymphocyte infiltration and PD-L1 expression, resulting in good clinical responses with anti PD-1/PD-L1 monoclonal antibodies, unlike cold tumors. Our hypothesis regarding cold tumors is to treat them with DC vaccines, which have been shown to increase lymphocyte infiltration and PD-L1 expression, and then, in a second step, to incorporate the anti-PD-1/PD-L1 monoclonal antibodies into the treatment in order to improve clinical responses.

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
