# Peer review of "Dendritic Cells in Cancer Immunology and Immunotherapy"

_cancers, 2024, doi:10.3390/cancers16050981_

Round 1

Reviewer 1 Report (Previous Reviewer 1)

Comments and Suggestions for Authors

The authors need to have the writing clear so that other investigators can understand it, as the authors will not be there to explain/ clarify the text to readers.

This problem may be overcome by them sending the manuscript to one of their colleagues who is fluent in English, as there are several layers of grammatical errors in this manuscript, which alters the scientific meaning of the text. Some are outlined below. Please address all areas below in blue.

Also importantly,  about 7 paragraphs under the title of hard selection of patients do not appear to be related to that topic.

Line 21: check point inhibitors (CPI) to chemotherapy has ….Please make check point one Word throughout the manuscript. Checkpoint.

Line 21. check point inhibitors (CPI) to chemotherapy has opened a new scenario in the treatment of cáncer …. Replace with  provided new options in ….

Line 29: improvements in vaccines formulations, selection of patients based on biomarkers

Line28. in the peripheral blood of patients without a significant clinical impact on outcome

Lines 35-37. such as interferon ((IFN) or APOBEC) and tumor microenvironment (TME) composition and function (tumor infiltrating lymphocytes TIL-and PD-L1/PD-1 axis) determine an inherent immunological status…  Too many “and”. Too many inappropriate brackets. Please rewrite this section.

Line 56. Furthermore, lack of immunogenicity…

Lines 56-59. Further, lack of immunogenicity in some tumors is due to a “cold” immune profile based on the lack of T cells infiltration, also due to a dysfunctional or exhausted T cell status in those “hot” tumors infiltrated by TIL that could be recovered by DCV. Please rephrase

Lines 65-67. In addition, this response could can generate immunological memory that would which may prevent relapses of the disease.

Lines 77-78. This is due to the fact that immature DC have low surface expression of chemokine receptors , and co-stimulatory molecules,

Line 82. cells produce inflammatory cytokines that translates into activation

Line 85 etc. A single tumor antigen or poorly immunogenic antigens used leads to a poor immunogenic DCV formulation. According to the source of tumor antigens employed, the induced immune response could change. Too many mistakes – sentence can’t be used. Please consider below.

 The use of a single tumor antigen or of poorly immunogenic antigens leads to a poor immunogenic DCV formulation. Depending on the source of tumor antigens employed, the resulting immune response may vary. Therefore, DC must be loaded with the  relevant tumor antigens.

Line 90. and disadvantages and for the momento currently, there are no data demonstrating

Line 94. it requires its characterization and this is a costly and time-consuming process.

Lines 102 etc. Although there is no consensus to establish the best route for DCV administration and knowing that it is possible to prime T cell immunity regardless of the route, the quality of responses may be different. Must rephrase for clarification.

Line 108. that hinder the antigen presentation to APC and T cell activation

Line 111: Regarding the subheading wrong selection of patients, it was already changed by May 2023 due to the suggestion of the reviewer, because the original title was a mistaken selection of patients. We put  Hard selection of patients.

This new sub-heading does not mean anything and cannot be used.

Please consider: Inappropriate selection of patients.

Lines 112 etc. Some of the negative results of DCV may also be due to the traditional tendency of early-phase clinical trials to include patients with advanced or heavily pretreated metastatic disease or associated comorbidities, or a powerful tumor-associated immunosuppression.
Must rephrase to make for clarification.

Line 115. Naïve of therapy ( Untreated patients with a strong immune response could respond better to all antitumoral strategies. Is this feasible? Cancer patients are usually so sick that they do need the initial chemotherapy for some basic level of disease control and then the DCV may be useful.  Please consider removing that sentence.

Line 116: clarification: we add to naïve of therapy (untreated); as we explained in our last answer to the reviewer, we use this expression commonly in the clinical arena.

Please do not use the term “naive of therapy patients”. It is not used in an English speaking Journal. If you mean “untreated” please use the word “untreated”.

Line 122. Since then, there has been a new renewed interest in the development of clinical

Line 130 etc. we improved the expression of the sentence. These cells produce high levels of IL-12p70, a molecule necessary for priming a TH1 response, than those generated with the previous protocol.       This sentence is not clear English. Please re-write. What previous protocol? State the cell type or whatever.

Line 136. The former ones are specialized in the recognition of viral antigens. Must state clearly which ones.

Line 145. The legend of Figure 1 has been clarified again. Schematic representation of the generation of DC vaccines. On the one hand, Monocytes isolated from peripheral blood can be differentiated into DCs after culture with GM-CSF and IL-4. On the other hand There are two main types of DCs in peripheral blood: pDCs (plasmacytoid)  and mDCs (myeloid). All these DCs must be loaded with peptides from tumor antigens, RNA encoding tumor antigens or tumor lysates to obtain the final product: mature and pulsed DCs for DCV therapy.

On the one hand

On the other hand – these two phrases are not acceptable. Please leave them both out.

All earlier text says DC rather than DCs. Be consistent.

Line 163. onto MHC-I molecules.  This process allows the activation of

Line 171: actually removed and added in fact –“in fact” is not aceptable for scientific writing.

In fact, Most likely, the combination of DCV with conventional/new treatments is mandatory to achieve synergistic effects.

Line 173 etc. In this way respect, the incorporation of the immune checkpoint inhibitors, (CPI) as one of the newest therapeutic strategies, has brought a revolutionized in the treatment of some solid tumors, and there is the potential of combining CPI with DCV as a useful therapy regimen. has opened up the possibility of combining both strategies.

Line 181 etc.  By In 1998, Fields described  reported an important antitumoral role of DC pulsed with tumor lysates in preclinical breast cancer (BC) models47 due to the activation of cytotoxic T lymphocytes (CTL)48, 49.         Please rephrase- due to- it does not fit there.

Line 186 etc. clinical benefit observed with fusion cells from BC with DCs in the metastatic scenario disease 52 was not seen in other studies53. Do not use the Word scenario here.

Line 188. Changing the methodology with DCV, an antiFoxP3+ Treg depletion immunotoxin was administered before giving viral modified DCV, improving immunogenicity in advanced BC patients54.  Please rephrase the whole sentence- Changing the methodology with DCV is not aceptable.

Line 194: Moving to the early clinical scenario, Qi et al treated early stage non-luminal BC patients with tumor lysate-pulsed DCV.

Moving to the early clinical scenario-  this phrase is not acceptable

Please consider- In a clinical trial with early stage disease ….

Line 200: with immunogenic activity both in the sentinel lymph nodes as well as in the systemic blood with a safety profile. Clarification: Safety profile means no adverse or mild toxicity with vaccines (fever, chills, dermal reactions in the DCV injection site); in summary a good tolerability profile to the vaccine.

…..with a safety profile?      Based on what the reference said consider using either: with minimal adverse effects,

 or with no adverse effects

Line 195. with tumor lysate-pulsed DCV showing benefit in 3-year PFS although there was no impact on OS

Clarification in previously line 201: Immunogenic response in the tumor means an increase rate of pathological complete responses as well as a rise in stromal TILs and PD-L1 expression in TNBC specimens (tumor and milieu) in the surgery versus the diagnostic biopsy after receiving DCV plus chemotherapy in the neoadjuvant scenario as compared to the control group without DCV. Immunogenic response in the peripheral blood reflects phenotypic changes in myeloid-derived suppressor cells, NK and T cells; increased blood cell proliferation and IFN-É£ production; a growth in humoral responses as well and changes in TCR-β repertoire after neoadjuvant combined treatment in BC patients (Santisteban M et al, Ther Adv Med Oncol 2021).

A little of the clarification above for immunogenic responses should be described in the text to give the reader clues concerning what the authors have in mind.

Line 200. as in the systemic blood with a safety profile56.  Rephrase in blue.

Line 200 etc. Our group reported that patients vaccinated with monocyte-derived autologous DC loaded with autologous tumor lysate experienced an increased tumoral immunogenic response in the tumor? Should the Word tumoral be left out?

Line 205. On the other hand, triple negative (TN) tumors have a grim prognosis but also are are also known…

Line 213: they have proved to be of no clinical benefit in immune desert tumors or in those tumors with dysfunctional or T cell exhausted, excluded or ignored scenarios.

Please do not use scenarios. Maybe phenotype can be used  instead of scenarios in this case here.

Line 222. Regarding toxicity, DCV present a mild one in monotherapy as well as in combination with radiation, chemotherapy and other biologic therapies in BC. Rephrase.

Regarding toxicity, DCV presents limited toxicity when used as monotherapy or in combination with radiation, chemotherapy or other biologic therapies in BC.

Line 232 etc. Therefore, chemoimmunotherapy combinations should look for? a more precised therapy for our patients as well as a chemotherapy de-escalation approach to avoid overtreatment of patients and unneeded toxicities.

chemoimmunotherapy combinations cannot look for – consider using the words, should utilize

Line 237: chemotherapy de-escalation approach to avoid overtreatment of patients and unneeded toxicities.  Remove unneeded, please consider unwanted toxicities.

Figure 2 text still needs to be further inproved as below.

Hot tumors are characterized by high lymphocyte infiltration and PD-L1 expression, obtaining resulting in good clinical responses with anti PD-1/PD-L1 monoclonal antibodies, On the other hand unlike cold tumors. do not meet these characteristics.

Line 250 etc. Diffuse gliomas constitute the most frequent malignant primary brain tumors in adults, particularly especially World Health Organization (WHO) grade 4 glioblastoma (GBM)77.

Line 262: explored in the past decade in such an immunologically “cold tumor” with (removed still) promising but not yet firm results

What are firm results?  This term can’t be used.     Maybe, conclusive results?

Line 267: Only a few clinical trials in this arena

Please do not use the word arena in any part of manuscript.

Do the authors mean: only a few GMB clinical trials are randomized studies and are designed to achieve a high level of efficacy?

Line 278.   … DCV were administered concomitantly with radiotherapy or chemotherapy. , but not in all of them.

Line 287 etc: Thus, autologous rather than standardized antigens-based vaccines might ensure a more personalized and better targeting of the full repertoire of antigens present on the patient’s tumor and might prevent from mutations of single targeted specific antigens.

….. and might prevent from mutations-  this term cannot be used

Line 299.
Adverse events are mild in severity and easily manageable.

Line 307. non-randomized phase I and II trials suggest a benefit on  may have survival benefit in patients with glioblastoma treated with…

Line 319: Randomized phase II trials revealed opposite (mixed) results, and a single (large) unique  randomized phase III trial did not allow us to draw a  conclusions regarding efficacy because of its….

Line 322. enrolled patients among such DCV trials on DCV and those whose

Line 348: but the humoral response was impaired.

Line 364: Clarification: observation is the same thing as no therapy.

Fifteen patients with disease-free resection margins were randomized 1:1 to receive DCV versus observation100. Replace observation with no therapy.

Line 368: a scarce (removed minimal)  benefit       There is no such thing as a “scarce benefit”   Please consider using, limited clinical benefit.

Line 373: One of the key limitations of DCV relies on is its inability to overcome

Line 375. to suppress Tregs have been driven  consider         conducted

Line 394. Solely cervical cancer…     Replace with        only

Line 407 etc. resulting in an impairment in antigen-presenting capabilities and number, even promoting cancer progression. Rephrase

resulting in an impairment in antigen-presenting cell (APC) function, reduced numbers of APC and promoting cancer progression.       Replace with this if above is trying to say this.

Line 408: However, DCV are safe and effective and provide to these gynaecological cancer patients a clinical improvement- improper grammar. Rephrase

Please consider: Provide clinical improvement for these gynecological patients…

Line 426. In order to prolong the lifespan of DC                where? In culture? In patients?

Line 430. which produced a CD4+ and cervical cancer-specific humoral immune response125.

What is a CD4+? Do the authors mean a CD4+ T cell .. Other cell types express CD4.

Line 431: Moving forwards on OC patients, clinical trials have established the safety of DCV, whereas effectiveness varies depending on production process, delivery and study design.
The term moving forward cannot be used.

Please consider: Clinical trials with ovarian cancer patients have established the safety of DCV, however, vaccine efficacy varies depending on the production process, route of delivery and study design.

Line 436. This fact was also related to higher levels of tumoral marker Ca 125 and a worse outcome in this population.

Line 437 etc. Nowadays Currently, more than 20 DCV clinical trials in OC have been are registered on ClinicalTrials.gov looking for designed to investigate new and efficient strategies against

Line 441. … vaccines to the SOC therapy providing could provide additional benefit as compared to SOC

Line 452. An increased PFS and a non-significant 16 months improval in OS was describe observed in OC patients

Line 456 etc. …. the potential combination of drugs to use with DCV or and the predictive/prognostic biomarkers used to select patients. Prospective cohort studies with large samples will provide the greatest evidence significant knowledge in the area of DC therapy in the future19.

Line 468 etc… more impressively importantly, produced objective tumor responses in 5/16 advanced melanoma patients133. A similar clinical trial with MAGE-3A1 peptide-pulsed monocyte derived DC also described resulted in the expansion of antigen-specific cytotoxic lymphocytes

Line 504. first of such clinical trials were reported in early 2000’s

Line 512. was reported to be safe and clinically active in a

Line 527. Authors reported that DFS at 24 months in the per-protocol population.

If above is not your group replace authors with     Others.

Line 550 etc. such as PSMA, MUC1, NY-ESO-1, MAGE and CDCA1, and some of them some of which have shown preliminary signals of activity.

Line 558: 21 patients with mCRPC were randomized to receive a vaccine with DC loaded with recombinant PSMA and survivin peptides or docetaxel and prednisone (SOC), presenting respectively response rates of 73% with DCV and 45% with chemotherapy respectively159.

Lines 578 etc. Regarding antigen selection, most clinical trials using defined antigens have been based on the use of TAA that which, despite

Line 587. has grown exponentially during the last years. … Replace with over the last several years….

Line 615. but during the last years have shown their….  Replace with,     more recently….

Line 627. MHC molecules in cancer correspond to these transcripts not corresponding to traditional mRNA molecules……  Please Rephrase

Line 632  etc. During the last years, More recently, CPI namely those targeting the PD-1/PD-L1 pathway, have become the backbone of most immunotherapy combinations182 Their impressive results in an important proportion of many hot, highly inflamed tumor types?

Line 636. On the other side Furthermore, the immunosuppressive tumor microenvironment

Lines 639 etc. release of immunosuppressive brakes imposed to tumor-specific T cells induced by vaccines would allow responses with stronger functional properties. Must rephrase.

Line 645. CPI would increase efficacy usually above that observed by these therapies when applied independently.

Line 648 etc: DCV have been able to induce detectable antitumor immune responses in cancer patients; but the magnitude or the context where this immunity has been primed, has not impacted sufficiently impacted on clinical outcomes in both either the early and or advanced scenarios in disease solid tumors.

Comments on the Quality of English Language

Needs much work.

Author Response

Dear reviewer,

thank you very much for your suggestions. We have incorporated most of them in the text and we have also required the english editing service of the journal in order to facilitate and clarify the understanding of the text. As you can see, we have attached the certificate of english editing.

We have also changed the subtitle about appropriate selection of the patients (instead of hard selection of patients).

As you have remarked, most of the content of this paragraph is not related to the topic. In the initial version of the paper this content was in the introduction in another section below . We have continued with this  text below the section 1.4 (not inside).

We agree with your regarding most of the suggestions. However, in the clinical arena, naïve of therapy is a pretty employed term in order to refer to untreated patients. We want to mantain this term (Line 123) with clarifications.

Regarding the manufacturing of DCV, we have incorporated the references that explain the initial protocol in anotehr sentence. In the Ref 25, you have this explanation in the abstract and in the methods section (Mailliard RB, Can Res 2004)

In line 310, the term referred to mutations in specific antigens is the one we want to mantain, and it si referred to genetic changes.

I hope all this efforts could help to improve the final version of the paper.

Best regards,

M. Santisteban, MD, PhD

Reviewer 2 Report (New Reviewer)

Comments and Suggestions for Authors

I did not participate to the review of the initial version. Thus my comment may appears inappropriate at this stage. My major comment relate to the structuration tumor type by tumor type, rather than by concept/strategy/type of DC/type of antigen,… the selected structuration prevent to really discuss the advantages/issues linked to each approach.

In the same line it will be important to address novelties in the field, there is a dedicated part on the neo-antigen which is nice, but in my opinion at least 2 critical parts are missing (that may be dispersed in the specific tumor sections): i) evidences that DCV turn cold tumors into hot tumors, ii) combined DCV treatment with immune checkpoints. This would clearly need to discuss mouse preclinical data and perspectives of clinical trials in human.

Other major concerns:

1-      Abstract should clearly state the content of the review and the elements that will be discussed

2-      CD34 derived DC as a sources of DC vaccine in clinical trial should be referred as an alternative option

3-      Many ref to reviews and not to the original papers

4-      the perspective at the end of the introduction part is an additional strangeness of this review

Comments on the Quality of English Language

no comments

Author Response

Dear reviewer,

I thank you for your suggestions regarding the last version of the paper you have received. The comment regarding structuration by tumor types is not going to be changed right now, but we will bear in mind for other communications.

We have just included a couple of sentences by the end of the abstract in order to summarize the content of the review based on your recommendation.

The evidence of transforming cold into hot tumors has been demonstrated by our group in breast cancer and has been comented in other relevant papers. References were cited (Refs 1; 2; 57; 58; 61; 76) in the paper and  has been also  suggested in other papers (Pfirschke C et al, Cell 2016; figure at the front page) not included in our paper.

To date, the combination of DCV with immune checkpoint inhibitors is a pretty new strategy that needs clinical results and evidence in order to be included in the text. To date and to our knowledge, only one clinical trial has been found in clinicaltrials.gov  (NCT05765084) in patients with malignant pleural mesothelioma treated with atezolizumab plus WT1/DCV that is under recruitment.

Regarding your comments about CD34 derived DC as a potential source of DC vaccines, this is a poor attractive option since the percentage of CD34 in the peripheral blood is pretty low (less than 1%). Even if we employ GM-CSF in order to obtain more CD34 in the bloodstream from bone marrow, the number that we obtained is very  low as compared to the number of monocytes, the main and elective way to obtain DCV.

Regarding the perspective at the end of the introduction, we think is a general explanation that could help to understand better the next sections regarding solid tumors and new trends.

Of course, editing english services of the journal have helped to improve English language.

Sincerely, I appreciate your kind comments,

Marta Santisteban, MD, PhD

Round 2

Reviewer 1 Report (Previous Reviewer 1)

Comments and Suggestions for Authors

Important
This is the previous review.

The authors need to complete the extensive changes requested from the last revision between pages 8-15 of this version. Please indicate changes in red font.

Additionally, in line 536-  38% (8/39) is inaccurate.   8/39= 20.5%

The authors need to have the writing clear so that other investigators can understand it, as the authors will not be there to explain/ clarify the text to readers.

This problem may be overcome by them sending the manuscript to one of their colleagues who is fluent in English, as there are several layers of grammatical errors in this manuscript, which alters the scientific meaning of the text. Some are outlined below. Please address all areas below in blue.

Also importantly,  about 7 paragraphs under the title of hard selection of patients do not appear to be related to that topic.

Line 21: check point inhibitors (CPI) to chemotherapy has ….Please make check point one Word throughout the manuscript. Checkpoint.

Line 21. check point inhibitors (CPI) to chemotherapy has opened a new scenario in the treatment of cáncer …. Replace with  provided new options in ….

Line 29: improvements in vaccines formulations, selection of patients based on biomarkers

Line28. in the peripheral blood of patients without a significant clinical impact on outcome

Lines 35-37. such as interferon ((IFN) or APOBEC) and tumor microenvironment (TME) composition and function (tumor infiltrating lymphocytes TIL-and PD-L1/PD-1 axis) determine an inherent immunological status…  Too many “and”. Too many inappropriate brackets. Please rewrite this section.

Line 56. Furthermore, lack of immunogenicity…

Lines 56-59. Further, lack of immunogenicity in some tumors is due to a “cold” immune profile based on the lack of T cells infiltration, also due to a dysfunctional or exhausted T cell status in those “hot” tumors infiltrated by TIL that could be recovered by DCV. Please rephrase

Lines 65-67. In addition, this response could can generate immunological memory that would which may prevent relapses of the disease.

Lines 77-78. This is due to the fact that immature DC have low surface expression of chemokine receptors , and co-stimulatory molecules,

Line 82. cells produce inflammatory cytokines that translates into activation

Line 85 etc. A single tumor antigen or poorly immunogenic antigens used leads to a poor immunogenic DCV formulation. According to the source of tumor antigens employed, the induced immune response could change. Too many mistakes – sentence can’t be used. Please consider below.

 The use of a single tumor antigen or of poorly immunogenic antigens leads to a poor immunogenic DCV formulation. Depending on the source of tumor antigens employed, the resulting immune response may vary. Therefore, DC must be loaded with the  relevant tumor antigens.

Line 90. and disadvantages and for the momento currently, there are no data demonstrating

Line 94. it requires its characterization and this is a costly and time-consuming process.

Lines 102 etc. Although there is no consensus to establish the best route for DCV administration and knowing that it is possible to prime T cell immunity regardless of the route, the quality of responses may be different. Must rephrase for clarification.

Line 108. that hinder the antigen presentation to APC and T cell activation

Line 111: Regarding the subheading wrong selection of patients, it was already changed by May 2023 due to the suggestion of the reviewer, because the original title was a mistaken selection of patients. We put  Hard selection of patients.

This new sub-heading does not mean anything and cannot be used.

Please consider: Inappropriate selection of patients.

Lines 112 etc. Some of the negative results of DCV may also be due to the traditional tendency of early-phase clinical trials to include patients with advanced or heavily pretreated metastatic disease or associated comorbidities, or a powerful tumor-associated immunosuppression.
Must rephrase to make for clarification.

Line 115. Naïve of therapy ( Untreated patients with a strong immune response could respond better to all antitumoral strategies. Is this feasible? Cancer patients are usually so sick that they do need the initial chemotherapy for some basic level of disease control and then the DCV may be useful.  Please consider removing that sentence.

Line 116: clarification: we add to naïve of therapy (untreated); as we explained in our last answer to the reviewer, we use this expression commonly in the clinical arena.

Please do not use the term “naive of therapy patients”. It is not used in an English speaking Journal. If you mean “untreated” please use the word “untreated”.

Line 122. Since then, there has been a new renewed interest in the development of clinical

Line 130 etc. we improved the expression of the sentence. These cells produce high levels of IL-12p70, a molecule necessary for priming a TH1 response, than those generated with the previous protocol.       This sentence is not clear English. Please re-write. What previous protocol? State the cell type or whatever.

Line 136. The former ones are specialized in the recognition of viral antigens. Must state clearly which ones.

Line 145. The legend of Figure 1 has been clarified again. Schematic representation of the generation of DC vaccines. On the one hand, Monocytes isolated from peripheral blood can be differentiated into DCs after culture with GM-CSF and IL-4. On the other hand There are two main types of DCs in peripheral blood: pDCs (plasmacytoid)  and mDCs (myeloid). All these DCs must be loaded with peptides from tumor antigens, RNA encoding tumor antigens or tumor lysates to obtain the final product: mature and pulsed DCs for DCV therapy.

On the one hand

On the other hand – these two phrases are not acceptable. Please leave them both out.

All earlier text says DC rather than DCs. Be consistent.

Line 163. onto MHC-I molecules.  This process allows the activation of

Line 171: actually removed and added in fact –“in fact” is not aceptable for scientific writing.

In fact, Most likely, the combination of DCV with conventional/new treatments is mandatory to achieve synergistic effects.

Line 173 etc. In this way respect, the incorporation of the immune checkpoint inhibitors, (CPI) as one of the newest therapeutic strategies, has brought a revolutionized in the treatment of some solid tumors, and there is the potential of combining CPI with DCV as a useful therapy regimen. has opened up the possibility of combining both strategies.

Line 181 etc.  By In 1998, Fields described  reported an important antitumoral role of DC pulsed with tumor lysates in preclinical breast cancer (BC) models47 due to the activation of cytotoxic T lymphocytes (CTL)48, 49.         Please rephrase- due to- it does not fit there.

Line 186 etc. clinical benefit observed with fusion cells from BC with DCs in the metastatic scenario disease 52 was not seen in other studies53. Do not use the Word scenario here.

Line 188. Changing the methodology with DCV, an antiFoxP3+ Treg depletion immunotoxin was administered before giving viral modified DCV, improving immunogenicity in advanced BC patients54.  Please rephrase the whole sentence- Changing the methodology with DCV is not aceptable.

Line 194: Moving to the early clinical scenario, Qi et al treated early stage non-luminal BC patients with tumor lysate-pulsed DCV.

Moving to the early clinical scenario-  this phrase is not acceptable

Please consider- In a clinical trial with early stage disease ….

Line 200: with immunogenic activity both in the sentinel lymph nodes as well as in the systemic blood with a safety profile. Clarification: Safety profile means no adverse or mild toxicity with vaccines (fever, chills, dermal reactions in the DCV injection site); in summary a good tolerability profile to the vaccine.

…..with a safety profile?      Based on what the reference said consider using either: with minimal adverse effects,

 or with no adverse effects

Line 195. with tumor lysate-pulsed DCV showing benefit in 3-year PFS although there was no impact on OS

Clarification in previously line 201: Immunogenic response in the tumor means an increase rate of pathological complete responses as well as a rise in stromal TILs and PD-L1 expression in TNBC specimens (tumor and milieu) in the surgery versus the diagnostic biopsy after receiving DCV plus chemotherapy in the neoadjuvant scenario as compared to the control group without DCV. Immunogenic response in the peripheral blood reflects phenotypic changes in myeloid-derived suppressor cells, NK and T cells; increased blood cell proliferation and IFN-É£ production; a growth in humoral responses as well and changes in TCR-β repertoire after neoadjuvant combined treatment in BC patients (Santisteban M et al, Ther Adv Med Oncol 2021).

A little of the clarification above for immunogenic responses should be described in the text to give the reader clues concerning what the authors have in mind.

Line 200. as in the systemic blood with a safety profile56.  Rephrase in blue.

Line 200 etc. Our group reported that patients vaccinated with monocyte-derived autologous DC loaded with autologous tumor lysate experienced an increased tumoral immunogenic response in the tumor? Should the Word tumoral be left out?

Line 205. On the other hand, triple negative (TN) tumors have a grim prognosis but also are are also known…

Line 213: they have proved to be of no clinical benefit in immune desert tumors or in those tumors with dysfunctional or T cell exhausted, excluded or ignored scenarios.

Please do not use scenarios. Maybe phenotype can be used  instead of scenarios in this case here.

Line 222. Regarding toxicity, DCV present a mild one in monotherapy as well as in combination with radiation, chemotherapy and other biologic therapies in BC. Rephrase.

Regarding toxicity, DCV presents limited toxicity when used as monotherapy or in combination with radiation, chemotherapy or other biologic therapies in BC.

Line 232 etc. Therefore, chemoimmunotherapy combinations should look for? a more precised therapy for our patients as well as a chemotherapy de-escalation approach to avoid overtreatment of patients and unneeded toxicities.

chemoimmunotherapy combinations cannot look for – consider using the words, should utilize

Line 237: chemotherapy de-escalation approach to avoid overtreatment of patients and unneeded toxicities.  Remove unneeded, please consider unwanted toxicities.

Figure 2 text still needs to be further inproved as below.

Hot tumors are characterized by high lymphocyte infiltration and PD-L1 expression, obtaining resulting in good clinical responses with anti PD-1/PD-L1 monoclonal antibodies, On the other hand unlike cold tumors. do not meet these characteristics.

Line 250 etc. Diffuse gliomas constitute the most frequent malignant primary brain tumors in adults, particularly especially World Health Organization (WHO) grade 4 glioblastoma (GBM)77.

Line 262: explored in the past decade in such an immunologically “cold tumor” with (removed still) promising but not yet firm results

What are firm results?  This term can’t be used.     Maybe, conclusive results?

Line 267: Only a few clinical trials in this arena

Please do not use the word arena in any part of manuscript.

Do the authors mean: only a few GMB clinical trials are randomized studies and are designed to achieve a high level of efficacy?

Line 278.   … DCV were administered concomitantly with radiotherapy or chemotherapy. , but not in all of them.

Line 287 etc: Thus, autologous rather than standardized antigens-based vaccines might ensure a more personalized and better targeting of the full repertoire of antigens present on the patient’s tumor and might prevent from mutations of single targeted specific antigens.

….. and might prevent from mutations-  this term cannot be used

Line 299.
Adverse events are mild in severity and easily manageable.

Line 307. non-randomized phase I and II trials suggest a benefit on  may have survival benefit in patients with glioblastoma treated with…

Line 319: Randomized phase II trials revealed opposite (mixed) results, and a single (large) unique  randomized phase III trial did not allow us to draw a  conclusions regarding efficacy because of its….

Line 322. enrolled patients among such DCV trials on DCV and those whose

Line 348: but the humoral response was impaired.

Line 364: Clarification: observation is the same thing as no therapy.

Fifteen patients with disease-free resection margins were randomized 1:1 to receive DCV versus observation100. Replace observation with no therapy.

Line 368: a scarce (removed minimal)  benefit       There is no such thing as a “scarce benefit”   Please consider using, limited clinical benefit.

Line 373: One of the key limitations of DCV relies on is its inability to overcome

Line 375. to suppress Tregs have been driven  consider         conducted

Line 394. Solely cervical cancer…     Replace with        only

Line 407 etc. resulting in an impairment in antigen-presenting capabilities and number, even promoting cancer progression. Rephrase

resulting in an impairment in antigen-presenting cell (APC) function, reduced numbers of APC and promoting cancer progression.       Replace with this if above is trying to say this.

Line 408: However, DCV are safe and effective and provide to these gynaecological cancer patients a clinical improvement- improper grammar. Rephrase

Please consider: Provide clinical improvement for these gynecological patients…

Line 426. In order to prolong the lifespan of DC                where? In culture? In patients?

Line 430. which produced a CD4+ and cervical cancer-specific humoral immune response125.

What is a CD4+? Do the authors mean a CD4+ T cell .. Other cell types express CD4.

Line 431: Moving forwards on OC patients, clinical trials have established the safety of DCV, whereas effectiveness varies depending on production process, delivery and study design.
The term moving forward cannot be used.

Please consider: Clinical trials with ovarian cancer patients have established the safety of DCV, however, vaccine efficacy varies depending on the production process, route of delivery and study design.

Line 436. This fact was also related to higher levels of tumoral marker Ca 125 and a worse outcome in this population.

Line 437 etc. Nowadays Currently, more than 20 DCV clinical trials in OC have been are registered on ClinicalTrials.gov looking for designed to investigate new and efficient strategies against

Line 441. … vaccines to the SOC therapy providing could provide additional benefit as compared to SOC

Line 452. An increased PFS and a non-significant 16 months improval in OS was describe observed in OC patients

Line 456 etc. …. the potential combination of drugs to use with DCV or and the predictive/prognostic biomarkers used to select patients. Prospective cohort studies with large samples will provide the greatest evidence significant knowledge in the area of DC therapy in the future19.

Line 468 etc… more impressively importantly, produced objective tumor responses in 5/16 advanced melanoma patients133. A similar clinical trial with MAGE-3A1 peptide-pulsed monocyte derived DC also described resulted in the expansion of antigen-specific cytotoxic lymphocytes

Line 504. first of such clinical trials were reported in early 2000’s

Line 512. was reported to be safe and clinically active in a

Line 527. Authors reported that DFS at 24 months in the per-protocol population.

If above is not your group replace authors with     Others.

Line 550 etc. such as PSMA, MUC1, NY-ESO-1, MAGE and CDCA1, and some of them some of which have shown preliminary signals of activity.

Line 558: 21 patients with mCRPC were randomized to receive a vaccine with DC loaded with recombinant PSMA and survivin peptides or docetaxel and prednisone (SOC), presenting respectively response rates of 73% with DCV and 45% with chemotherapy respectively159.

Lines 578 etc. Regarding antigen selection, most clinical trials using defined antigens have been based on the use of TAA that which, despite

Line 587. has grown exponentially during the last years. … Replace with over the last several years….

Line 615. but during the last years have shown their….  Replace with,     more recently….

Line 627. MHC molecules in cancer correspond to these transcripts not corresponding to traditional mRNA molecules……  Please Rephrase

Line 632  etc. During the last years, More recently, CPI namely those targeting the PD-1/PD-L1 pathway, have become the backbone of most immunotherapy combinations182 Their impressive results in an important proportion of many hot, highly inflamed tumor types?

Line 636. On the other side Furthermore, the immunosuppressive tumor microenvironment

Lines 639 etc. release of immunosuppressive brakes imposed to tumor-specific T cells induced by vaccines would allow responses with stronger functional properties. Must rephrase.

Line 645. CPI would increase efficacy usually above that observed by these therapies when applied independently.

Line 648 etc: DCV have been able to induce detectable antitumor immune responses in cancer patients; but the magnitude or the context where this immunity has been primed, has not impacted sufficiently impacted on clinical outcomes in both either the early and or advanced scenarios in disease solid tumors.

Comments on the Quality of English Language

Not acceptable.

Author Response

Dear reviewer,

I am pretty concerned because all the changes you told me to implement the paper have been already done. I am not sure you are reading the last version of the text. The control changes tool has been employed fot this purpose. 

Regarding your last new comment, in line 536, we said "In a phase 2 single arm trial in combination with ipilimumab that included 39 stage III-IV melanoma patients, a remarkable response rate of 38% (8/39 were complete responses) was reported" and of course 8/39 are the complete responses and 7/39 are the partial responses; 15 out of 39 patients refers to this  38%. Our text is  accurate because 38% was the total of partial plus complete responses. 

English text has been improved by your editing service following your recommendations, thats why I thought you have not checked our last version of the text that I attach again. 

As we told in our last submission, the paragraph under "Inappropiate selection of patients" is independent, so we have a visual separation within the text. 

Please do not hesitate to contact us if needed.

Marta Santisteban

Reviewer 2 Report (New Reviewer)

Comments and Suggestions for Authors

none

Author Response

Responses are explained, completed and enhanced in red font with the right changes. Most of the comments are not academic but based on English language. By November 10th, we answered to academic and scientific comments; and we applied the tool control changes to modify language style in the text after the English edition services were requested to you and incorporated to the paper. We have added all the changes an answered point by point to all the grammatical, linguistic mistakes and suggestions following your recommendations. We have been very surprised by the need to incorporate to the reviewers point by point not only scientific suggestions but also linguistic ones, because time is precious for all of us and English edition was applied into the text.

We think this version is very suitable for the journal.

-The first comment related to line 525 is completely accurate, and we do not agree with the reviewer, because we said “In a phase 2 single arm trial in combination with ipilimumab that included 39 stage III-IV melanoma patients, a remarkable response rate of 38% (8/39 were complete responses) was reported”. Clarification: total responses (ORR) and complete responses are different endpoints, the 38% data is referred to total responses (ORR= partial + complete responses) and in our written comment, eight of 39 patients got complete responses, and 7 of 39 acquired partial responses; thus 15 of 39 patients refers to this 38% of ORR result.

-The manuscript has been sent to the editorial service of the journal Cancers in order to improve the English language, and the text has been corrected following their recommendations.

-The paragraph under the title “1.4. Inappropriate selection of patients” (lines 113 to 120) is related to the subheading. In a separate paragraph (lines 124 to 186) we have commented the rationale to support that DCV could be an interesting and a promising approach nowadays. Reviewer comments said that the text (lines 124 to 186) do not appear to be related to that topic, however in this paragraph we explain the strengths of DCV, and we consider it should be placed in this introduction section after the text related to disadvantages and weaknesses showed with DCV.

-Line 21 has been changed to checkpoint inhibitors instead of checkpoint…

-Lines 21-22 have been replaced to “Although the addition of checkpoint inhibitors (CPI) to chemotherapy has provided new options in the treatment of cancer….”

-Line 28 has been changed to “ un the peripheral blood of patients without a significant clinical impact on outcome…..”

-In line 29 we have added the plural form like this “improvements in vaccines formulations….”

-We have rewritten the paragraph from lines 34 to 37 without less brackets and “and” with this final result: “Patient characteristics (such as age, genetics, microbiome, previous infections or exposure to immune modified drugs), as well as tumor features (tumor mutational burden -TMB-, microsatellite instability -MSI-, mismatch repair deficiency -dMMR-, specific immune signatures such as interferon or APOBEC and tumor microenvironment….”. Probably multiple abbreviations have also concurred in this paragraph as well, and we have tried to improve it.

-Line 57 was improved and instead of further we have included “furthermore”……

-We have rephrased the paragraph in lines 57 to 60 with this result: “Furthermore, lack of immunogenicity in some “cold” tumors based on the absence of T cells infiltration, but also due to a dysfunctional or exhausted T cell status in those inflamed tumors, could be recovered by DCV.

-Lines 66-67 have been update to “In addition, this response can (instead of could)  generate immunological memory which may (instead of that would) prevent relapses of the disease”.

-Line 79 was changed to “This is due to the fact that immature DC have low surface expression of chemokine receptors, and co-stimulatory molecules….”

-Line 83 was modified to “Differentiated T cells produce inflammatory cytokines that translates…..”

-Line 86 to 89 have been substituted by “The use of a single tumor antigen or poorly immunogenic antigens leads to a poor immunogenic DCV formulation. Depending on the source of tumor antigens employed, the resulting immune response may vary. Therefore, DC must be loaded with the relevant tumor antigens..….” as you suggested, instead of “A single tumor antigen or poorly immunogenic  antigens used leads to a poor immunogenic DCV formulation. According to the source of tumor antigens employed, the induced immune response could change.”

-Lines 92-93 have been improved with “They all have advantages and disadvantages and for the moment currently….”

-Line 96: “However, it requires its characterization and this is a costly and time…..”

-We have rephrased lines 105-106 from “although there is no consensus to establish the best route of DCV administration and knowing that it is possible to prime T cell immunity regardless of the route, the quality of responses may be different” by “although there is no consensus to establish the best route for DCV administration, the quality of responses may be different”

-Line 110: that hinder the antigen presentation

-We have changed the subheading Inappropriate selection of patients instead of Hard selection of patients (the first one was Mistaken selection of patients) in line 113.

-Lines 114 to 116 have been rephrased to Some of the negative results of DCV may also be due to the trend of early-phase clinical trials to include patients with advanced or heavily pretreated metastatic disease. These situations are linked to a tumor-associated immunosuppression. The initial sentence was “Some of the negative results of DCV may also be due to the traditional tendency of early-phase clinical trials to include patients with advanced or heavily pretreated metastatic disease or associated comorbidities, or a powerful tumor-associated immunosuppression.

-Line 117: However, therapy-naïve (newly diagnosed and untreated) oncologic patients…..The explanation in brackets has been added as suggested by the reviewer in a previous version of the paper. This reviewer asks if this situation is feasible (the untreated scenario, I suppose) since cancer patients are usually very sick and need therapy. Naïve of therapy refers to patients that have received no therapy for their disease. The most usual explanation is because they have been recently diagnosed from the disease and they have not started therapy yet; but also severe comorbidities, active COVID19 infection, … could justify the lack of therapy. Here, the point is that multiple previous therapies are associated to a weaker immune response and that therapy-naïve patients have a stronger immune system. In fact; I have removed the text in brackets as in the first version, because I think it is confusing. In the clinical scenario we are very used to employ the therapy-naïve term. The reviewer suggests to remove naïve of therapy term and to use untreated, however the therapy-naïve term is used in plenty of academic publications (check pubmed) and it reflects exactly what I want to communicate. I understand each reviewer has his/her own style; my understanding is this not an essential point.

-Line 128: I have added “Since then, there has been a renewed (instead of new) interest in the development….” as suggested

-Line 137 refers to previous protocol that is very common and described in other paper (reference 25) that we have just cited.

In lines 136-138, we have changed the previous sentence by this newer one “These cells produce high levels of IL-12p70, a molecule necessary for priming a TH1 response, than those generated with the previous protocol25-27.”

-Legend of Figure 1 has been modified for clarification: “Schematic representation of the generation of DC vaccines. On the one hand, Monocytes isolated from peripheral blood can be differentiated into DC after culture with GM-CSF and IL-4. On the other hand, Moreover, there are two types of DC in peripheral blood: pDC (plasmacytoid)and mDC (myeloid). All these DC must be loaded with peptides from tumor antigens, RNA encoding tumor antigens or tumor lysates to obtain the final product: mature and pulsed DCs.” We have removed the expressions on the one hand and on the other hand as suggested.

-Line 168. We added in the text: Cross-presentation is the presentation of extracellular antigens (which are usually bound to MHC-II molecules) onto MHC-I molecules: this process allows the activation of CD8+ T cells, which are essential in the antitumor response.

-Line 177 we have removed in fact because is not acceptable for scientific writing and we have put “Most likely, the combination of DCV with conventional…..”

-Lines 179-182: “In this way, the incorporation of the immune checkpoint inhibitors (CPI) as one of the newest therapeutic strategies has brought a revolution in the treatment of some solid tumors and there is the potential of combining CPI with DCV as a useful therapy regimen”……instead of has opened up the possibility of combining both strategies.

-We have rephrased due to in the lines 188-190 “By 1998, Fields described an important antitumoral role of DC pulsed with tumor lysates in preclinical breast cancer (BC) models47 driven by (instead of due to) the activation of cytotoxic T lymphocytes (CTL)48, 49.

-Line 194 should removed the word scenario here: “However, clinical benefit observed with fusion cells from BC with DCs in the metastatic disease (instead of scenario)……..

-Line 195, we have changed Changing the methodology with DCV by “Improving DCV procedure…….”

-Line 201 modified Moving to the early clinical scenario by “In a non-randomized clinical trial for early disease….”

-Line 206, changes suggesting “with immunogenic activity both in the sentinel lymph nodes as well as in the systemic blood with no adverse effects56 instead of with a safety profile. Already changed.

-Reviewer want us to incorporate in the text a clarification answered previously. The clarification we added is explained in the referenced paper (REF 57) generated from our group. Although we think is not necessary, we add an explanatory paragraph: “Our group reported that patients vaccinated with monocyte-derived autologous DC loaded with autologous tumor lysate experienced an increased immunogenic response in the tumor (increased pCR); its milieu (a rise in TIL and PD-L1 expression) and the peripheral blood (described as phenotypic changes in myeloid-derived suppressor cells, NK and T cells, increased blood lymphocytes proliferation and IFN-É£ production, a growth in humoral responses as well as changes in TCR-β repertoire) when we compare diagnostic specimens with surgical specimens after chemoimmunotherapy, although no dramatic changes in survival were found in non-overexpressing HER2 early BC”. We also left out tumoral immunogenic responses in line 209.

-Line 216: “tumors have a grim prognosis but are also known to respond better….”. I am not sure what the reviewer wants since in the comments he/she wrotes: “but also are are also known….”

-Lines 223-225: “they have proved to be of no clinical benefit in immune desert tumors or in those tumors with dysfunctional or T cell exhausted, excluded or ignored phenotypes (instead of scenarios)”

-Lines 223-224, rephrased to “Regarding toxicity, DCV presents limited toxicity when used as monotherapy or in combination with radiation or other biologic therapies in BC” instead of Regarding toxicity, DCV present a mild one when used as monotherapy as well as in combination with radiation, chemotherapy or other biologic therapies in BC.

-Line 244 modified to “Therefore, chemoimmunotherapy combinations should become (instead of look for) a more precised therapy”

-Line 246: “chemotherapy de-escalation approach to avoid overtreatment of patients and unwanted toxicities” instead of unneeded.

-Legend of Figure 2 changed to “Hot tumors are characterized by high lymphocyte infiltration and PD-L1 expression, resulting in (obtaining) good clinical responses with anti PD-1/PD-L1 monoclonal antibodies. (REMOVED On the other hand,) unlike cold tumors (REMOVED do not meet these characteristics).

-Line 261: changed to “particularly World Health Organization (WHO) grade 4 glioblastoma” instead of especially

-Line 273: changed to “with promising but not yet consistent results.” Instead of firm

-Lines 277-278 rephrased and clarified to “Only a few GMB clinical trials are randomized studies and are designed to achieve a high level of efficacy” instead of only a few clinical trials in this arena

-Line 288 removed “DCV were administered concomitantly with radiotherapy or chemotherapy, but not in all of them.”

-Lines 296-299 said “Thus, autologous rather than standardized antigens-based vaccines might ensure a more personalized and better targeting of the full repertoire of antigens present on the patient’s tumor and might avoid mutations of single targeted specific antigens. “ Reviewer asks not to use the term might prevent from mutations; I do not know what he/she wants us to modify; the mutations in the antigens are related to resistance to antigen-specific therapy. DCV are autologous and multiantigen vaccines, so there are able to target multiple antigens expressed in the tumor with higher antitumoral responses. We are opened to your suggestions.

-Line 308 was modified to “Adverse events are mild in severity (added to the text) and easily manageable”

-Line 316: ….phase I and II trials suggest a benefit on survival in patients with glioblastoma: The reviewer puts “phase I and II trials suggest a benefit on may have survival benefit” I do not understand which change she/he wants.

-Line 325 was changed to “Randomized phase II trials revealed mixed results….” (instead of opposite.

-Line 331 puts “enrolled patients among such DCV trials” instead of patients among such trials on DCV.

-Line 357: “but the (added) humoral response….”

-Line 372: “Fifteen patients with disease-free resection margins were randomized 1:1 to receive DCV versus no therapy…”. The reviewer suggests to modify observation by no therapy, although we already explained to her/him that the term has the same meaning and it is very used in the clinic.

-Line 377: “a limited clinical benefit” instead of scarce; this comment was modified by August in our text

-Line 382: “One of the key limitations of DCV relies on its inability…” instead of is

-Line 384: “to suppress Tregs have been conducted…” instead of driven

-Line 404: “Only cervical cancer” was placed instead of Solely

-Line 417:“resulting in an impairment in antigen-presenting cells (APC) function, reduced numbers of APC, (instead of antigen presenting capabilities and number) and (instead of even) promoting cancer progression” ….

-Line 418 has been improved to: “DCV are safe and effective and provide to these gynaecological cancer patients a clinical improvement” instead of DCV are safe and effective and provide to these gynaecological cancer patients a clinical improvement

-Line 436: “In order to prolong the lifespan of DC and delay apoptosis, siRNA can be added to the HPV antigens124. To clarify, these results have been observed in preclinical and clinical studies.

-Lines 438-440: Clarification. “These patients were subcutaneously injected with a DCV carrying keyhole limpet haemocyanin (KLH) and full-length HPV16/18 E7, which produced a CD4+ and cervical cancer-specific humoral immune response”. CD4+ is a surface membrane glycoprotein. It is expressed in T helper lymphocytes, but also in monocytes, macrophages and dendritic cells. It works as a co-receptor of TCR with an APC and interacts directly with MHC-II in order to acquire a cytotoxic and antitumoral activity.

-Lines 441-443: “Moving forwards on OC patients, clinical trials have established the safety of DCV, whereas effectiveness varies depending on production process, delivery and study design”. As suggested, we had already changed the sentence in August.

-Line 446: change was already performed in August to “This fact (added) was also related to higher levels of tumoral marker Ca 125”

-Lines 447-449: “Currently, more than 20 DCV clinical trials in OC have been registered” instead of Nowadays

-Line 451: “vaccines to the SOC therapy providing additional benefit” was changed to could provide

-Line 461: “An increased PFS and a non-significant 16 months improval in OS was observed in OC…” instead of described

-Line 466: “the potential combination of drugs to use (added) with DCV, or the predictive/prognostic biomarkers used to select patients. Prospective cohort studies with large samples will provide the greatest evidence in the future (instead of significant knowledge in the area of DC therapy) already performed by August

-Line 482: “more impressively, produced objective tumor responses in 5/16 advanced melanoma”. Reviewer put ……more impressively importantly ….so I do not know what she/he wants, since both words are underlined in the same color, and previous changes in the text have been incorporated before. This issue has been repeated frequently.

-Line 484: “A similar clinical trial with MAGE-3A1 peptide-pulsed monocyte derived DC also resulted in the expansion” instead of described

-Line 518:” first of such clinical trials were reported (added)

-Line 526: “was reported to be safe and clinically active” was added

-Line 543: Others reported that DFS at 24 months in the per-protocol population instead of authors because it is not our group

-Line 567: “Other DCV under development for prostate cancer include different antigens such as PSMA, MUC1, NY-ESO-1, MAGE andCDCA1, etc, and some of which (instead of them) have shown preliminary signals of activity

-Line568-570: 21 patients with mCRPC were randomized to receive a vaccine with DC loaded with recombinant PSMA and survivin peptides or docetaxel and prednisone (SOC), presenting (respectively removed) response rates of 73% with DCV and 45% with chemotherapy respectively159.

-Line 598: “based on the use of TAA which, despite selective expression….” instead of that

-Line 605: “the identification of tumor-specific mutations has grown exponentially over the last several years…..” instead of during the last years

-Line 634:” but during the last years have shown their capacity to be translated into proteins…..” has been replaced by more recently

-Line 649: “MHC molecules in cancer correspond to these transcripts not related to traditional mRNA molecules” has been changed to instead of corresponding

-Line 654: changes from During the last years to “More recently, CPI, namely those targeting the PD-1/PD-L1 pathway…”

-Line 656-657: “Their impressive results in an important proportion of hot, highly inflamed tumors types with elevated TMB levels….” (addition)

-Line 658: “Furthermore, the immunosuppressive tumor microenvironment has been proposed….” instead of On the other side

-Lines 660-662: “Therefore, release of immunosuppressive brakes imposed to tumor-specific T cells induced by vaccines would allow responses with stronger functional properties.” I think this sentence is correct; if we eliminate the immunosuppressive milieu, stronger antitumoral responses could be obtained.

-Line 667: “combination of vaccines (generating a new pool of tumor-specific T cells) together with CPI would increase efficacy above that observed by these therapies….” instead of usually

-Lines 670-673: “DCV have been able to induce detectable antitumor immune responses in cancer patients; but the magnitude or the context where this immunity has been primed, has not impacted sufficiently on clinical outcomes in both the early and advanced scenarios in (disease removed) solid tumors. Already done.

I hope this version would be acceptable for you. Sadly, I would not know what else to do if you do not approve this version.

Round 3

Reviewer 1 Report (Previous Reviewer 1)

Comments and Suggestions for Authors

Reviewer Comments

The authors have made several improvements to this manuscript but many other changes need to be implemented.
They should check cross references and see that all literature changes suggested here are what the original authors intended to say.

The authors should change all grammar and incorrect word changes stated in this revision before further consideration/ review of this manuscript.

ABSTRACT: Cancer immunotherapy modulates the immune system by avoiding?? immune escape….
This sentence is not accurate. It seems more accurate that tumors act/ survive by immune escape, whereas cancer immunotherapy overcomes immune escape by tumors.  

e.g. Article  Front Med 2022 Apr;16(2):208-215. doi: 10.1007/s11684-022-0922-5. Epub 2022 Apr 4. Multi-target combinatory strategy to overcome tumor immune escape Yingyan Yu

Line 29.     Thus, improvements in vaccines formulations….

Line 30    ….. with other antitumoral therapies are needed to enhance patients survival.

Line 58  …………… of T cells infiltration, but also

Line 89   ……….. with the relevant tumor antigens. DC must be loaded with relevant tumor antigens.     Repetition

Line 92      ………. and disadvantages and for the moment currently, there are no data

Line 118 ……a strong immune system could may respond better to all antitumoral strategies.

 Line 124     Regarding all Due to these disadvantages, 

Line 136      generating mature but no exhausted a−type-1 polarized DC

Line 154   are two main types of DC in peripheral blood.    
Depending on the classification used others may report more types of DC in peripheral blood.

Line 154.   All these DC must be loaded…..

Line 156. to obtain the final product: mature and pulsed DCs

Be consistent with the use of DC or DCs throughout the article. It is easier to leave off the s on DCs for the whole article.

Line 195. ….. with fusion cells from BC with DCs in the metastatic

 Line 196…….. To improve Improving DCV procedure therapy, an anti-

Lines 208-216.  Our group reported that patients vaccinated with monocyte-derived autologous DC loaded with autologous tumor lysate experienced an increased immunogenic response in the tumor (increased pCR); its milieu (a rise in TIL and PD-L1 expression) and the peripheral blood (described as phenotypic changes in myeloid-derived suppressor cells, NK and T cells, increased blood lymphocytes proliferation and IFN-É£ production, a growth in humoral responses as well as changes in TCR-β repertoire) when we compare diagnostic specimens with surgical specimens after chemoimmunotherapy, although no dramatic changes in survival were found in non-overexpressing HER2 early BC.

This sentence is too long and is not clear. A sentence cannot be 7 lines.
Please state where pCR is earlier defined.
Please ensure that TIL is defined.

Line 221. TIL and PD-L1 expression as compared to more advanced scenarios disease.
In standard English the word scenario is not used in a science journal.

Line 224. both in the early and the advanced arena disease.
Arena is not an acceptable word to use.

Line 235. Regarding toxicity, DCV presents limited toxicity when used as monotherapy

Lines 246…..
Therefore, chemoimmunotherapy combinations should become a more precised may be a more effective therapy for our patients. With this strategy as well as  a chemotherapy de-escalation approach may be used to avoid overtreatment of patients and unwanted toxicities.

Line 311. DCV for GBM have shown an excellent safety and tolerability profiles across in all clinical trials.

Line 312. Adverse events are mild in severity and easily manageable.

Line 315. disease itself or to other used concomitant therapies. therapies used concmmitantly.

Line 316. no relevant adverse events nor or toxicity attributable

Lines 329-332.   This statement is not acceptable please consider below in blue if it describes the article referred to.… and only a unique single randomized phase III trial did not allow to draw a conclusion as to efficacy because of its cross-over design with nearly 90% of patients in the control group receiving DCV after recurrence.
These words are redundant. Only =1. Unique=1. Single =1.

… and a single randomized phase III trial was inconclusive because of its cross-over design with nearly 90% of patients in the control group receiving DCV after recurrence.

Line 334. However major differences in methodological design and enrolled patients among in such trial DCV trials and those whose data were used as external controls for comparison constitute a major 336 limitation and preclude from drawing such firm conclusions90, 91 .

Lines 336-337. and those whose data were used as external controls for comparison constitute a present major limitations and preclude from drawing such firm conclusions therefore valid conclusions cannot be drawn.

Line 381……  Potential areas of improvement include…..

Line 423..
However, DCV are safe and effective and provide to these gynaecological cancer patients a clinical improvement provide clinical improvement to these gynaecological cancer patients.

Line 445. produced a CD4+ and cervical cancer-specific humoral immune response

Do the authors mean a CD4+ T cell as other cell types express CD4.

Line 446. Moving forwards on In OC patients, clinical trials have established

Line 451. This fact was also related to higher levels of tumoral marker Ca 125 and a worse outcome in this population
The above sentence has no meaning Here and can be deleted.

Line 457. The increased potential benefit of moDC vaccination in addition to SOC

Line 460. In the a second phase II clinical trial

Lines 470…..
Therefore, these findings show that DCV have additional  limited benefit. , but also reveal that . However, further research is required at the level of the DCV manufacturing, the potential combination of drugs with DCV, and/ or with the predictive/prognostic biomarkers used to select patients to enable further improvement in DCV therapy.

Line 481. approaches have tried to targeted this immune cell population in melanoma patients

Line 490. DC, suggested showed a higher capacity of the cell-free preparation in terms of antigen-specific

Line 493….. This finding raises the key unanswered questions on regarding the factors that influence the induction of an adequate T cells priming in patients. Such knowledge allows us to better identify patients in need of DCV to enable them to mount an effective immune response against the tumor.

Lines 497…..
Many years later, the   Recent technological advances allowed a more individualized approach in to antigen selection. Personalized MHC-restricted neoantigen peptide-pulsed ex vivo induced DC were able to amplify existing antigen-sepcific T-cell responses but could also produce responses to other antigen that were not detectable prior to vaccination. This was a phase 1 trial that which included only three patients with resected stage III melanoma, a design that precluded antitumor response evaluation.
The third sentence may need to be altered and come before the second sentence. This section does not read well.
Please consider: a phase 1 trial that included only three patients with resected stage III melanoma was conducted (for sentence 2). In any case Rephrase this section.

Line 519…. Butterfield et al. developed several clinical trials with DCV in melanoma patients. The first of such clinical trials were reported

Line 544. DFS at 24 months in the per-protocol population was significantly prolonged
Is DFS defined in the text?

Line 569. For example, in a small trial, 21 patients with mCRPC were randomized to receive a vaccine with DC loaded with recombinant PSMA and survivin peptides or docetaxel and prednisone (SOC). presenting Cohorts showed response rates of 73% with DCV and 45% with chemotherapy respectively.

Line 677-8. to identify individuals with tumors lacking TIL, or with exhausted or dysfunctional TIL, that would benefit from immune-priming strategies.

Comments on the Quality of English Language

All comments are detailed in the full report.

Author Response

Good morning,

I attach the corrections regarding our paper Dendritic cells in cancer immunology and immunotherapy following your recommendations. I have also uploaded the last version of the paper. 

Marta Santisteban, MD, PhD

This manuscript is a resubmission of an earlier submission. The following is a list of the peer review reports and author responses from that submission.

Round 1

Reviewer 1 Report

Comments and Suggestions for Authors

Strengths: The authors summarize the status of dendritic cell vaccines in several cancers. This area of translational medicine has had great anticipation of success and hence it is interesting to prepare a literature update in this field. The authors presented an extensive list of references.

Points of concern: There are several areas which need to be improved in this manuscript.

1. The first sentence of the introduction, lines 32-37 is not well structured and as such loses its meaning.

2. Lines 66-93 are not well written. Points 1-4 are not well outlined/ described. The authors should make specific points and then expand on them.

3. Figure 1 legend- lines 120-121. The authors need to show evidence that pDC are loaded with peptides for therapy. References? Typically pDC are regarded as regulatory harmful DC in cancer.

4. Mature alpha-polarized DC-1 have been used in several clinical trials and should be discussed in the text.

5. Several cancers have been discussed in which the authors outline the use of DC vaccines in clinical trials. However the efficacy of DC vaccines in these trials is not often stated/ described.

6. The conclusion section is not satisfactory. It is more like a brief summary.

7. Please define abbreviations before use e.g. Lines 401 and 407 SOC.

8. Lines 195-196 need references. In the same way, DCV upregulates expression of the PD-1/PD-L1 axis as an adaptive resistance mechanism due to the increased of TILs within breast tumors.

9. Lines 270- GBM section. State how many patients were used in the clinical trial.

10. Line 148 reference 32- This appears to be a prostate cancer clinical trial rather than a breast cancer model as mentioned in the text.

11. Overall there are grammar errors throughout which significantly alters the meaning of the text.

e.g.  Then, DC mature and present efficiently these peptides. Line 42.

Lines 140-145 should be rewritten. In this sense, the appearance of antibodies that act at the level of immune check points, which have brought about a revolution in the treatment of some solid tumors, has opened up the possibility of combining both strategies. …

Lines 184-189 need to be re-written.

Lines 195- 200 and lines 231-232, need to be re-phrased.

Line 344- Immune defeating properties?  Maybe immune suppressive properties.

Strengths: The authors summarize the status of dendritic cell vaccines in several cancers. This area of translational medicine has had great anticipation of success and hence it is interesting to prepare a literature update in this field. The authors presented an extensive list of references.

Points of concern: There are several areas which need to be improved in this manuscript.

1. The first sentence of the introduction, lines 32-37 is not well structured and as such loses its meaning.

2. Lines 66-93 are not well written. Points 1-4 are not well outlined/ described. The authors should make specific points and then expand on them.

3. Figure 1 legend- lines 120-121. The authors need to show evidence that pDC are loaded with peptides for therapy. References? Typically pDC are regarded as regulatory harmful DC in cancer.

4. Mature alpha-polarized DC-1 have been used in several clinical trials and should be discussed in the text.

5. Several cancers have been discussed in which the authors outline the use of DC vaccines in clinical trials. However the efficacy of DC vaccines in these trials is not often stated/ described.

6. The conclusion section is not satisfactory. It is more like a brief summary.

7. Please define abbreviations before use e.g. Lines 401 and 407 SOC.

8. Lines 195-196 need references. In the same way, DCV upregulates expression of the PD-1/PD-L1 axis as an adaptive resistance mechanism due to the increased of TILs within breast tumors.

9. Lines 270- GBM section. State how many patients were used in the clinical trial.

10. Line 148 reference 32- This appears to be a prostate cancer clinical trial rather than a breast cancer model as mentioned in the text.

11. Overall there are grammar errors throughout which significantly alters the meaning of the text.

e.g.  Then, DC mature and present efficiently these peptides. Line 42.

Lines 140-145 should be rewritten. In this sense, the appearance of antibodies that act at the level of immune check points, which have brought about a revolution in the treatment of some solid tumors, has opened up the possibility of combining both strategies. …

Lines 184-189 need to be re-written.

Lines 195- 200 and lines 231-232, need to be re-phrased.

Line 344- Immune defeating properties?  Maybe immune suppressive properties.

Author Response

Thank you very much for the comments regarding our paper "Dendritic cells in cancer immunology and immunotherapy".

-We have changed the first sentence in the introduction to improve its meaning

-We have clarified and explained lines 66-93 in a more extended text outlining different parts (lines 98-192 in the new text)

-We have changed the legend of figure 1  and we have add a reference regarding pDC loading with antigens/peptides for therapy 

-We have discussed about DC-1 in the text (lines 206-211)

-Some of the clinical trials mentioned in the text have not concluded its results yet; thus the efficacy of DCV in these cases is not described

-We have improved the conclusion section in the abstract as well as in the text

-In the GBM section, we had the number of patients included in our study (32). The number of patients included in this trial is low, but the number of patients included in the randomized trials is between 13 and 232. In the review performed by Datsi A et al, Front Immunol 2021, you can check these data on Table 1. The biggest study is the one published by Liau LM et al in JAMA Oncol 2023 (REF 64).

-SOC in lines 401 and 407 means standard-of-care as described previously in line 291 in the colorectal cancer section

-In line 148, we have added the correct reference from Fields et al, PNAS 1998

-We have re-written and re-phrased the text (lines 42, 140-145, 184-189, 195-200, 231-232, 344,.....) as you suggested, and we have also included a new melanoma section

-We have included the reference 71 from Sharma P et al, Cell 2017, to the text

Thank you again for your comments because they have helped to improve the final result.

Reviewer 2 Report

Comments and Suggestions for Authors

The authors have presented a significant review article deciphering the potential of  Dendritic Cells in Cancer Immunology and Immunotherapy. The article has been framed well but there are a few sections, where considerable changes are needed. The quality of the diagrams is appropriate. However, author can add a table summarizing the efficacy of natural products in this context. All sections are well correlated to each other. If possible, the author can add a paragraph mentioning the methodology being followed to prepare this review article which would further enhance the reader's attention. Citations are enough but try to incorporate the last 10 years' citations more in your article. Hence, I would suggest a few changes:

1. The conclusion portion of the abstract is not well presented and of which the article will not gain readers' attention.

2. Try to incorporate a separate section mentioning different therapeutic approaches related to the use of dendritic cells using plant-based compounds or nanotechnology.

3. Conclusion section is not at all acceptable. Kindly rewrite it. It should be able to explain the application of your review article in cancer therapeutics. 

Author Response

We deeply thank your comments regarding the paper entitled "Dendritic cells in cancer immunology and immunotherapy". So far, we have modified our first manuscript version by:

-improving conclusions in both the abstract and the manuscript sections, connecting them with the whole text

-our methodology has been based on the best and recent papers (only two recent abstracts) related to DCV and cancer; as well as high quality clinical trials chosen by the authors in their area of expertise. Of course we have included few pionering but older studies. The search has been performed in  PubMed and in Web of Science databases. Since this review is not a systematic review, we think there is no need to describe it in depth.

-we have not considered the use of plant-based and other natural compounds since it is a very broad subject. However, we have incorporate a new melanoma  section, since it is a high immunogenic tumor in which most types of immunotherapy have helped to improve patients  outcome and QoL.

We hope you are going to find a higher quality paper than in your first revision.

Thank you so much again,

Reviewer 3 Report

Comments and Suggestions for Authors

Comment #1. To make the manuscript more effective, it may be helpful to clarify its unique contributions and demonstrate how it offers advantages and novelty compared to existing literature on dendritic cells and immunotherapy in the context of cancer. This could help position the manuscript within the field and highlight its relevance. By doing so, the authors can make a more compelling case for why this review is valuable and important for advancing our understanding of the role of dendritic cells in cancer immunotherapy.

Comment #2. The authors should provide a more detailed description of the role of dendritic cells in cancer immunology, as indicated in the title of the manuscript. While the introduction briefly mentions the role of DCs as antigen-presenting cells, it is important to provide a more comprehensive overview of their functions and mechanisms in the context of cancer. Additionally, the statement that there is a decrease in the number and function of DCs in cancer patients (lines 50-51) should be supported by more recent and robust literature references. As such, the authors should revise this section to ensure that the evidence presented is accurate, up-to-date, and supported by strong scientific sources.

Comment #3. Before describing the results of dendritic cell vaccination in various tumors, the authors should provide a more detailed explanation of the rationale behind this approach. This will ensure that readers have a better understanding of how DC vaccination works and why it is being used as a therapeutic strategy. By providing a comprehensive overview of the mechanisms underlying DC vaccination, the authors can set the stage for a more in-depth analysis of the results obtained in different tumor types. Part of this information could be found in the introduction section, which would benefit from being split into smaller subsections to better organize and present the information. This will help to ensure that the reader can more easily follow the flow of the argument and identify the key points being made.

Comment #4. Could you please provide further clarification on lines 90-94, particularly with regard to the fourth item? It is unclear what this item refers to, and additional information is needed to better understand the context and significance of this statement.

Comment #5. It could be valuable to include information on the use of GM-CSF and IFN-alpha, in addition to IL-4 and GM-CSF, for dendritic cell differentiation. The work of Caterina Lapenta has highlighted the importance of these factors in promoting DC differentiation and function. Including this information in the discussion of DC differentiation would make the manuscript more comprehensive.

Comment #6. It is important to include information on dendritic cell vaccination in other tumor types that have been extensively studied in both published and ongoing trials. These include melanoma, lung, pancreatic, and non-solid hematological tumors. By providing a comprehensive overview of the use of DC vaccination in a variety of tumor types, the authors can help to highlight the potential benefits and limitations of this therapeutic approach. This will also help to contextualize the results obtained in the specific tumor types that are the focus of this review and allow for a more nuanced discussion of the overall efficacy and safety of DC vaccination in cancer treatment.

Comment #7. To facilitate comparison and better understand the main features of the studies discussed for each type of tumor, it would be valuable to include a table summarizing key information such as study design, patient characteristics, intervention type, and outcomes. This would enable readers to identify similarities and differences quickly and easily between the studies, as well as provide a clear overview of the main findings. By including a table, the authors can enhance the clarity and accessibility of their review, making it more informative and useful for researchers, clinicians, and other stakeholders in the field.

Comment #8. Given the complex and dynamic nature of the tumor microenvironment (TME) and its potential impact on immune responses, it would be interesting to consider whether the TME varies between different types of tumors and whether this variability could affect the response to dendritic cell vaccination differently. Although this issue is not explicitly addressed in the manuscript, it is an important topic to consider. Further research is needed to fully understand how the TME influences the efficacy of DC-based immunotherapies and to identify strategies that can optimize the response in different tumor contexts.

Comment #9. It is worth noting that the review does not cover certain relevant topics that could complement and expand on the discussion of dendritic cell vaccination in cancer immunotherapy. For instance, two notable omissions are the use of immunogenic cell death to prepare immunogenic tumor cell lysates and the concept of in situ dendritic cell vaccination. These approaches have gained attention as promising strategies to enhance the immunogenicity of tumor cells and induce potent antitumor immune responses. In the interest of providing a comprehensive overview of the field, future iterations of the review could consider incorporating these topics to provide a more holistic perspective on dendritic cell-based immunotherapies.

Author Response

We deeply thank your comments regarding the paper entitled "Dendritic cells in cancer immunology and immunotherapy". According to your recommendations, we have incorporated some changes to improve the final version of the paper:

-regarding comment #1, we have underlined the importance of this active cell therapy from our clinical experience. As explained in the abstract and in the text, some cold tumors, or those with dysfunctional or exhausted T cells are not able to respond to CPI and need DCV in order to reorganize the TME and to generate an adequate immune response.

-we have expanded and organized the introduction in subsections in order to provide a more comprehensive overview of DC functions, mechanism of action and applications; as well as specific tumor topics such as melanoma. We also have add new references in the text to support scientific contents, and clarify specific sentences as suggested in comment #2, #3,#4 and #5.

-regarding comment #6, we have incorporated a melanoma section.

-however, a table in which all tumors are put together does not seem a good option for us since it would be pretty long, complex and heterogeneous;  and it is not specific for one tumor type; as mentioned in comment #7. 

-tumor microenvironment (TME) has been  cited in some specific tumors . In fact, is the rationale shown in  Figure 2. All the comments regarding TIL in the text according to each tumor type are referred to this issue.

-we have added some concepts regarding the route of administration of the DCV as well as the types of lysates (lines 112-199)

We hope these points have served to improve our work.

Thank you so much again for these accurate suggestions.

Round 2

Reviewer 1 Report

Comments and Suggestions for Authors

1.This manuscript is greatly improved, however there are grammatical errors and poor sentence structure in many places. Below are some examples.

Line 27  therapy has demonstrated the ability to modify the tumor microenvironment to immune

Line 66   the lack of T cells infiltration, but or also due to a dysfunctional or exhausted T cell status in

Lines 79, 80 and 81, 82 are repeated

Line 90 However, mature DC express higher levels of all these structures both MHC- 90 II and co-stimulatory molecules

Line 92 Differentiate T cells

Line 93  produce inflammatory citokines that translates                    spelling

The poor choice of the antigens employed (a

  Line 95   A poor choice of antigen

Line 97 imply  results in a poor immunogenic DCV formulation.

Lines 102-103                             although it seems  clear?  that loading DC with MHC-I and II epitopes could promote the quality of the immune response generated. Add a reference to indicate that it seems clear

Lines 106 and 108-  The word identification does not seem to make sense –Maybe use characterization instead ?

Lines 108-112- A reference is required

clinical trials could may? not be the most suitable for achieving the migration of DC to the lymph       117

A mistaken selection of patients 124    Wrong choice of patients

or associated comorbidities, including powerful tumor-associated immunosuppression 127
This in blue is not a co-morbidity

Naïve of therapy patients  129  This does not make sense-  maybe untreated?

Line 131  As a consequence,    Of what?

check points  line 189     This is one word

with tirosin kinase inhibitors (TKI) in 421  spelling

Lines 460-462  Is not clear.
Furthermore, more  than 20 DCV clinical trials in OC have been registered on ClinicalTrials.gov, with the objective of focusing on the most fatal tumor in which there is the greatest research on DC1

those genes encoding classical mRNA molecules that originate? proteins annotated in pro-   Line 637

Lines 672-675 need to be revised

2.Figure 2 is not clear. One group of cells is cold tumor, another is hot tumor. What is the third group of cells? The panel at the bottom of Figure 2 is not sufficient for full explanation. More colors should be used and the receptors on the groups of cells should be labeled.  
The groups of cells need to be readily understood visually by the reader. Even with the panel/ code at the bottom.

Author Response

Thank you very much for your comments, rewiever 1.

We have made a big effort to amend  the grammatical errors, to add new references (refs 20 and 21) and to improve the Figure 2.

The only thing we would like to preserve is the naïve of therapy (untreated) population, as we name it in our clinical arena.

We are pretty convinced this last version has improved very much and we are deeply grateful to you.

Do not hesitate to contact us if needed.

Reviewer 2 Report

Comments and Suggestions for Authors

The authors have incorporated all the suggested changes in an appropriate manner.

Citations are also improved.

The language of the manuscript has been thoroughly revised.

The conclusion section is well-revised.

Hence the manuscript can be accepted in its current form 

Author Response

Thank you very much to rewiever 2.

Round 3

Reviewer 1 Report

Comments and Suggestions for Authors

REVIEW
This is a detailed review article however the authors need to clarify several aspects of this manuscript as mentioned below, and recheck any related clinical trials data.

The number at the end of the text refers to the line.

they have shown proved no clinical benefit in immune desert tumors    22

but or also due to a dysfunctional or exhausted T cell status in those “hot” tumors infil- 58

this response generates immunological memory that may prevents relapses of the disease. 66
There is no evidence that the immunological memory generated by DC vaccines in patients prevents relapses of the disease- so the corrected statement above is more accurate.

 tors, and co-stimulatory molecules, and do not secrete cytokines that are mandatory for T-cell 77

or poorly immunogenic antigens used imply results in a poor 83
or poorly immunogenic antigens used leads to a poor ----- 83

Starting at line 104: The oral route is related to may have anatomical barriers such as gastrointestinal tract pH acid milieu and digestive enzymes that hinder the antigen presentation to APC and T cell activation.

knowing  whereas? these are not the best patients for this kind of strategy. Naïve of therapy patients
Line 113

Line 115 onwards has nothing to do with the wrong selection of patients.
A new paragraph and new sub-heading must start there.

Regarding all these disadvantages, Clinical progress in DCV stalled for years, until     Line 115

could delay disease impairment . Line 119
What is disease impairment? State clearly if it improved overall survival.

These cells produce higher levels of IL-12p70, a molecule necessary for priming    128

than the DCs generated with the previous protocol (TNF- a, IL-1b, IL-6 and    129

Figure 1. Schematic representation of the generation of DC vaccines. On the one hand, Monocytes isolated from peripheral blood can be differentiated into DCs after culture with GM-CSF and IL-4. On the other hand, There are two main types of DCs in peripheral blood: pDCs and mDCs. All these DCs must be loaded with peptides from tumor antigens, RNA encoding tumor antigens or tumor lysates to obtain the final mature DC product for DCV. mature and pulsed DCs.

Although the collected       line 164

clear therapeutic results are only achieved in less than 15% of patients.  166

progress should be made in optimizing the use of this treatment (Figure 1). Actually, 169

Actually The combination of DCV with conventional/new treatments may be mandatory necessary to achieve ---- 170

In this way, the incorporation of the immune checkpoint inhibitors (CPI) 171 as one of the newest therapeutic strategies has brought a revolution in the treatment of 172 some solid tumors and has opened up the possibility of combining both strategies.
Line 171 etc
Therapy such as immune checkpoint inhibitors (CPI) have been very effective in the treatment of some solid cancers and combination therapy of CPI and DCV may be a promising treatment strategy for improved outcome.

Line 174 etc
Therefore, optimizing both the design of the vaccine formulation as well as the design 174 of clinical trials, the clinical efficacy of DCV could be improved.
Therefore, the clinical efficacy of DCV may be improved by optimizing the design of the vaccine formulation as well as testing DCV in combination with other therapy in clinical trials.

 2. RELEVANT SOLID TUMORS
Why relevant? All are relevant.

Consider
2. DC VACCINES IN SOLID TUMORS

clinical benefit observed with fusion cells from BC with DCs in the advanced scenario  line 184
The term advanced scenario is not clear? Do the authors mean advanced disease patients?

Changing the methodology with DCV, an anti- 185 FoxP3+ Treg depletion immunotoxin was administered before giving viral modified DCV, 186 improving immunogenicity in advanced BC patients54 .

The administration of an anti-FoxP3+ Treg depletion immunotoxin before giving a viral modified DCV was found to improve immunogenicity in advanced BC patients ( )    Line 185, 186.

showing and demonstrated a clinical benefit in up to one third of the     189

Moving to the early clinical scenario, Qi et al treated early stage non-luminal BC patients with tumor lysate-pulsed DCV. These patients showed benefit in 3-year PFS although there was no impact on OS. In this study, almost 60% of the patients showing immune delayed hypersensitivity responses (55).    Line 193

Line 196-7    well as in the systemic blood with a safety profile.
What does with a safety profile mean?

Line 197 etc
Our group reported that patients vaccinated with monocyte-derived autologous DC loaded with autologous tumor lysate experienced an increased tumoral immunogenic response in the tumor, its milieu and the peripheral blood, although no dramatic changes in survival were found in non-overexpressing HER2 early BC.
What are these immunogenic responses?

Line 202 etc
On the other hand, Triple negative (TN) tumors have a grim prognosis but also are  known to respond better than other BC biologic subtypes to immunotherapy, based on higher TIL levels, more PD-L1/PD-1 activation, increased TMB and an increased neoantigen (neoAg) production (59-63). Regarding this issue, Early TNBC stands out as having increased TIL and PD-L1 expression as compared to more advanced scenarios disease 64. Although the addition of check point inhibitors (CPI) to chemotherapy may offer a new landscape new treatment options in the treatment of TNBC patients, both in the early (63, 65-68) and the advanced arena  stage (69-72) mainly due to a better outcome, CPI have proved to  be of no clinical benefit in immune desert tumors or in those tumors with dysfunctional or exhausted T cells.  excluded or ignored scenarios 61. In this way, DCV have demonstrated the ability to modify the tumor microenvironment and to potentiate systemic host immune responses as an active approach to treat BC patients by increasing T cell infiltration within the tumor (58) but also increasing PD-L1 expression in tumoral cells and stimulating systemic T cell activity against BC (57) .

excluded or ignored scenarios 
This is not clear. Excluded may already be an immune desert.
Why are DCV being compared with CPI in the paragraph above in italics?

Regarding toxicity, DCV shows mild toxicity when used as present a mild one in monotherapy as well as or in combination with radiation, chemotherapy and other biologic therapies in BC. Combination of DCV with chemotherapeutic agents such as cyclophosphamide, gemcitabine or temozolamide (74) increases toxicity but also shows improved clinical responses by mechanisms such as triggering immunogenic cell death, modifying the permeability of CD8 T cells to cytolytic factors and decreasing Tregs and myeloid-derived suppressor cells (MDSC).

Radiation therapy upregulates class I MHC class I and enhances tumor-cross presentation in BC (75) .
What does this mean?

In the same way, DCV upregulates expression of the PD-1/PD-L1 axis as an adaptive resistance mechanism due to an increased infiltration of TILs within breast tumors (76). In fact, this transformation from cold into hot immune tumors could increase the target population allow more patients to benefit from CPI after DCV together used in combination with chemotherapy (Figure 2). Therefore, chemo-immunotherapy combinations should look for a more precised therapy for our patients as well as a chemotherapy de-escalation approach to avoid overtreatment of patients and unneeded increased toxicities.

Figure 2.

Last part….

Our hypothesis for the latter ones cold tumors is to treat them with DCs vaccines, which have been shown to increase lymphocyte infiltration and PD-L1 expression, and then add anti-PD-1/PD-L1 monoclonal antibodies to the treatment, to achieve better clinical responses.  thus achieving good clinical re- 243 sponse. 244 2.2.

2.2. BRAIN TUMORS

line 257
extensively explored in the past decade in such an immunologically “cold tumor” with still promising but not yet firm results.

Early studies in animal models of GBM showed that DCV triggered a specific anti-tumor immune response and resulted in oncolytic activity? with tumor growth reduction and prolonged survival, leading to the translation of this strategy to the clinic through phase I-II trials. Only few of them, including a phase III trial??, are randomized studies attempting to provide a higher level of evidence in terms of efficacy (79 ). Patient populations included in these clinical trials has been heterogeneous, with either newly diagnosed, recurrent GBM or even both groups of patients.
The above text needs to be clarified. What is oncoltytic activity?

and might prevent form mutations of single targeted specific antigens.  285   Please clarify

Line 294
Overall, DCV for GBM has shown an excellent safety and tolerability profile across

all in most clinical trials. Adverse events are mild in severity and easily manageable. The most

trial no relevant adverse events nor toxicity attributable to the immunotherapy were     299

Starting at line 312
Randomized phase II trials revealed mixed results84-88, and no conclusions regarding efficacy could be drawn in a

large unique randomized phase III trial did not allow to draw a conclusion as to efficacy

because of its cross-over design with nearly 90% of patients in the control group receiving

DCV after recurrence89.

However major differences in

methodological design and enrolled patients enrolled among such trial on in the DCV trial and those whose

data were used as external controls for comparison constitute a major limitation and pre-

clude from drawing such firm hence conclusions could not be drawn 90, 91. 

COLORECTAL CANCER
With more than 600.000 deaths estimated each year
What is 600.000?

Anti-EGFR or anti-VEGF

Line 339
Different routes of DCV administration were based on pre-clinical models showing that i.v. de-

livery provides a better anti-tumour response against visceral metastases (such as?), whereas non-

visceral metastases (such as ?) respond better to the intradermal route. KLH-specific T-cell responses

Line 356
Fifteen patients with disease-free resection margins were randomized 1:1
to receive DCV versus observation100.
What is observation? No treatment?

Line 361   safe and only mild side effects were recorded101. Although immune responses have been

generated following DCV, a scarce minimal clinical benefit has been shown in all these trials.

Line 368
properties of the tumor-microenvironment. Therefore, treatment combinations of DCV

with prior cyclophosphamide in order to suppress Tregs have been driven investigated  105.

line 402
However, DCV are safe and effective and may provide clinical improvement to

these gynaecological cancer patients a clinical improvement 119.

Line 415
 It has been observed that the concentration numbers of DC

is low in cervical tumours, while that of Treg cells is high in cervical cancer lesions, a phenomenom which

may be significantly associated with the persistence of hrHPV (Need to ADD A REFERENCE).

Is there a difference between cervical tumours and cervical cancer lesions?

hrHPV-  Is this earlier defined?

Line 425
Moving forwards on OC patients, In ovarian cancer clinical trials, have established the safety of DCV has been established, but there has been limited efficacy.

whereas effectiveness varies depending on production process, delivery and study de-

sign.

Line 429 etc Below is not a rationale
The rationale for incorporating DCV to the clinic was the discovery of a lower number of

DC in patients with OC than in healthy donors. This fact was also related to higher levels

of tumoral marker Ca 125 and a worse outcome in this population126.

Line 431
Nowadays Currently, more than 20 DCV clinical trials in OC have been registered on ClinicalTrials.gov looking for                  

Line 439
 In the second phase II clinical trial, stage III OC patients were treated with moDC vaccine either concurrently with chemotherapy, sequentially, or with chemotherapy alone.
This is not clear.

Line 451 Prospective cohort studies with large samples will

need to provide the greatest evidence in the future19.

This has no meaning.

MELANOMA
Line 469

Interestingly, a publication in 2005 reported that the presence of an-

tigen-specific lymphocytes in delayed-type hypersensitivity test skin biopsies predicted

progression-free survival in patients treated with a variety of DCV protocols22.
Did this predict longer/ better progression free survival?

Line 502
not included in the vaccine was described147, as well as additional clinical responses in a

Line 523
received DCV (62,9 vs. 34,8%). However, results from another randomized placebo-con-
State 62.9 vs 34.8%

Line 544 etc   This is not clear.
Other DCV under development for prostate cancer include different antigens such

as PSMA, MUC1, NY-ESO-1, MAGE and CDCA1, etc, and some of them which have shown prelim-

inary signals of activity. For example, in a small trial, 21 patients with mCRPC were ran-

domized to receive a vaccine with DC loaded with recombinant PSMA and survivin pep-

tides or docetaxel and prednisone, presenting respectively response rates of 73% and

45% (159). 

Line 630 etc   The text in italics is not clear
On the other side, The immunosuppressive tumor microenvironment has been proposed as one of the potential causes of potentially explaining the poor effect of vaccines despite their immunogenicity. Therefore, release of immunosuppressive brakes imposed to tumor-specific T cells induced by vaccines would allow responses with stronger functional properties.  634

4. CONCLUSION

DCV have been able to induce detectable antitumor immune responses in cancer pa-

tients; but the magnitude or the context where this immunity has been primed, has not

impacted sufficiently on clinical outcomes in both the early and advanced scenarios in

solid tumors.       This statement needs to be clarified.

Line 657
as CPI, have to be explored, in order to find synergistic effects between DC vaccination and